# Structural basis for saxitoxin congener binding and neutralization by anuran saxiphilins

Sandra Zakrzewska [1], Samantha A. Nixon [1], Zhou Chen [1,10], Holly S. Hajare[2], Elizabeth R. Park [2], John V. Mulcahy [2], Kandis M. Arlinghaus[3], Eduard Neu [4], Kirill Konovalov[4], Davide Provasi[4], Tod A. Leighfield[3], Marta Filizola [4], J. Du Bois [2] & Daniel L. Minor Jr. [1,5,6,7,8,9] ✉

Dinoflagellates and cyanobacteria produce saxitoxin (STX) and ~50 congeners that disrupt bioelectrical signals by blocking voltage-gated sodium channels ($Na_V$s). Consuming seafood carrying these toxins causes paralytic shellfish poisoning (PSP). Although $Na_V$s and anuran STX binding proteins (saxiphilins, Sxphs) use convergent STX binding modes, the structural basis for STX congener recognition is unknown. Here, we show that American bullfrog (*Rana catesbeiana*) *Rc*Sxph and High Himalaya frog (*Nanorana parkeri*) *Np*Sxph sequester STX congeners using a 'lock and key' mode shared with STX. Importantly, functional studies demonstrate that Sxph 'toxin sponges' reverse $Na_V$ block by multiple STX congeners and detect these toxins in a radioligand binding assay (RBA) used for environmental testing. Together, our study establishes how Sxphs sequester select neurotoxins and uncover STX congener-specific interactions distinct from $Na_V$s. These findings expand understanding of toxin sponge action and provide a foundation for strategies to monitor and mitigate the harmful effects of STX congeners.

Saxitoxin (STX) is the archetype of a family of lethal, paralytic, small-molecule guanidinium neurotoxins that block voltage-gated sodium channels ($Na_V$s) and their bioelectrical signals in nerve and muscle[1–3] and is a chemical weapon[2,3]. Diverse species of dinoflagellates and cyanobacteria in marine and fresh water harmful algal blooms produce this toxin class as cocktails of STX and STX congeners, some of which match or exceed STX toxicity[1,4–6]. Accumulation of STX and related toxins in seafood causes paralytic shellfish poisoning (PSP) and poses an increasing threat to commercial fishing and public health from harmful algal blooms[2,7,8]. Consequently, there are extensive worldwide efforts to detect contamination of food and water supplies by PSP toxins[9,10]. Structural studies of STX bound to $Na_V$s[11,12] and frog saxiphilins (Sxphs)[13,14], a class of soluble high affinity STX binding proteins[13,15], reveal a common STX recognition mode[13,14] by which these two structurally and functionally unrelated proteins use a relatively rigid binding site to coordinate the di-cationic, bis-guanidinium core of the toxin through a similar set of ionic and cation-π interactions[13,14]. How STX congeners bind to these targets and whether differences in binding affinity arise from changes to the STX binding mode or by some other means is unknown.

[1]Cardiovascular Research Institute, University of California, San Francisco, CA, USA. [2]Department of Chemistry, Stanford University, Stanford, CA, USA. [3]National Oceanic and Atmospheric Administration, National Centers for Coastal Ocean Science, Charleston, SC, USA. [4]Department of Pharmacological Sciences, Ichan School of Medicine at Mount Sinai, New York, NY, USA. [5]Department of Biochemistry and Biophysics, University of California, San Francisco, CA, USA. [6]Department of Cellular and Molecular Pharmacology, University of California, San Francisco, CA, USA. [7]California Institute for Quantitative Biomedical Research, University of California, San Francisco, CA, USA. [8]Kavli Institute for Fundamental Neuroscience, University of California, San Francisco, CA, USA. [9]Molecular Biophysics and Integrated Bio-imaging Division, Lawrence Berkeley National Laboratory, Berkeley, CA, USA. [10]Present address: Department of Anatomy and Physiology, Shanghai Jiao Tong University School of Medicine, Shanghai, China. ✉e-mail: daniel.minor@ucsf.edu

Natural modifications to STX yield a family of ~50 STX congeners, collectively known as paralytic shellfish toxins (PSTs)[1,2,5], that affect toxin affinity for Na$_V$s[2,3] and frog plasma containing Sxph activity[15]. STX is commonly modified at three positions (Fig. 1A): R1, the carbamoyl site; R2, the six membered guanidinium ring N1 position; and R3, the C11 carbon. R1 modifications include acylation (acetate, mono- and bis-hydroxybenzoate), decarbamoylation (e.g., decarbamoyl STX, dcSTX) and sulfation (e.g., gonyautoxin 5, GTX5, also known as B1). R2 hydroxylation creates neosaxitoxin, neoSTX, a more effective Na$_V$ blocker[16] and more potent poison[1,5] than STX. R3 can be modified by carboxymethylation (e.g., 11-saxitoxin ethanoic acid) and sulfation (e.g., gonyautoxin 2/3, GTX2/3), creating two interconverting epimers[1–3]. Combinations of these alterations further diversify the STX family. The chemical complexity and variable proportions of STX congeners in natural PST samples poses challenges for quantitative testing[17] and for developing PSP treatments.

Anuran Sxphs from diverse frogs and toads comprise a family of soluble, transferrin-like 'toxin sponge' proteins that contain a single high-affinity (Kd ~ nM) STX binding site[13–15]. This property allows Sxphs to rescue Na$_V$s from STX block[13,18] and likely contributes to the ability of some frogs to resist STX poisoning[15,18–20]. Competitive radioligand binding studies of bullfrog[15] and cane toad plasma[21] suggest that Sxphs can bind select STX congeners. However, there have been no direct studies of STX congener binding to purified Sxphs and no structural

information exists for STX congener binding to either Sxphs[13,14] or Na$_V$s[11,12], yielding a gap in understanding STX congener recognition.

Here, we investigate how STX congeners interact with their molecular targets by characterizing toxin binding to RcSxph from the American bullfrog (Rana catesbeiana)[13,14] and NpSxph from the High Himalaya frog (Nanorana parkeri)[13], two previously characterized Sxphs that share a common toxin binding pocket but that have different STX affinities[13]. Our studies using a suite of thermofluor (TF), fluorescence polarization (FP), and isothermal titration calorimetry (ITC) binding assays[13,22] show that both Sxphs have similar selectivity profiles ranging from enhanced binding of GTX5 to very weak affinity for the neoSTX series. High resolution X-ray crystal structures of NpSxph:STX congener complexes together with molecular dynamics (MD) simulations reveal toxin binding poses identical to STX and identify a network of ordered water molecules involved in toxin recognition. These findings highlight the ability of the pre-organized, relatively rigid STX binding pocket to accommodate many STX modifications and uncover structural differences that affect the binding preferences of Sxphs and Na$_V$s for different STX congeners. Importantly, functional studies using two-electrode voltage clamp (TEVC) and planar patch clamp experiments establish that Sxphs act as 'toxin sponges'[18] that can reverse Na$_V$ block by various STX congeners. Further, we demonstrate that Sxphs can substitute for rat brain homogenates in a widely used radioligand receptor binding assay (RBA) for

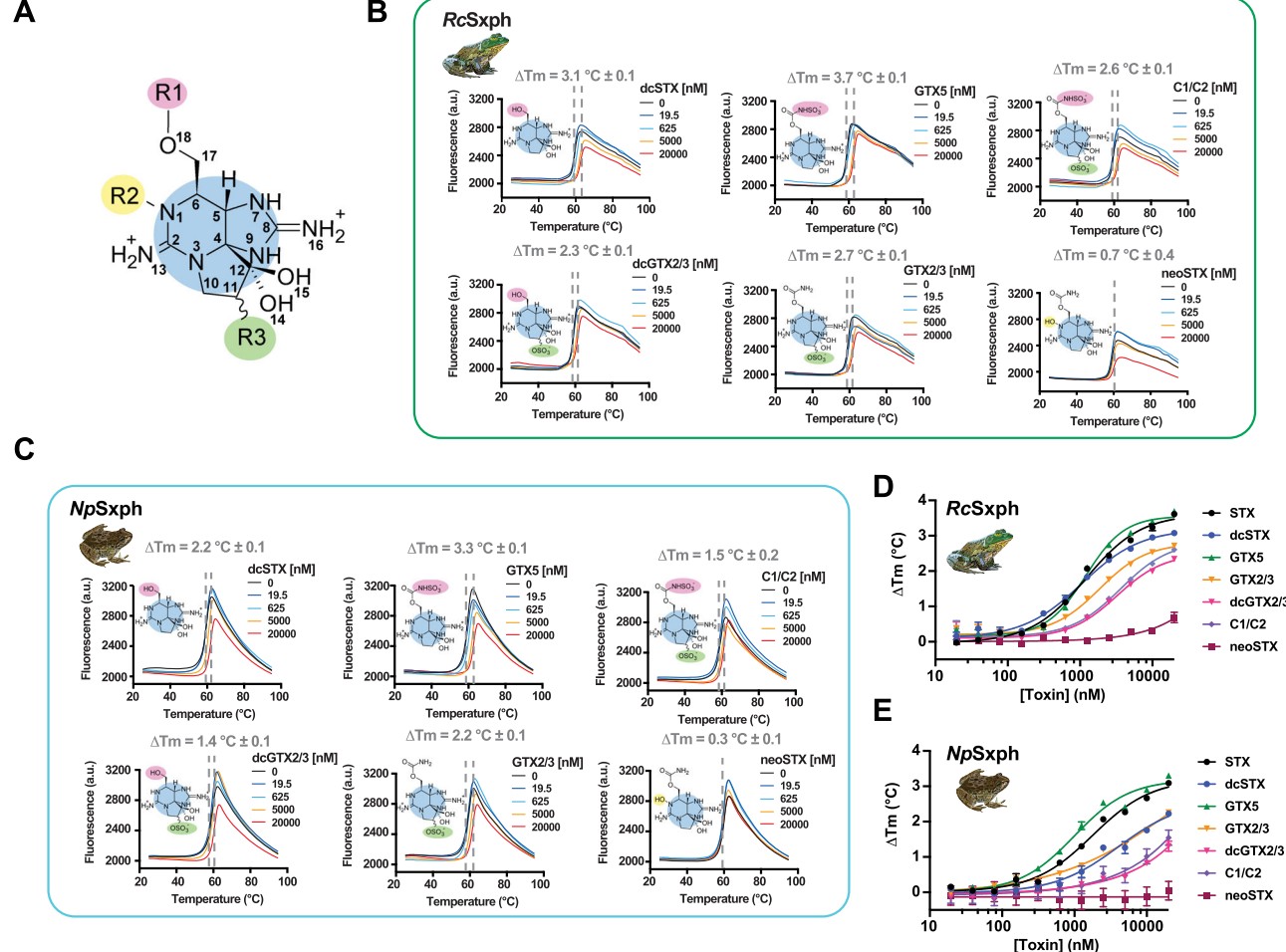

**Fig. 1 | TF assays reveal Sxph gonyautoxin binding preferences. A** STX core showing modification sites R1 (magenta), R2 (yellow), and R3 (green). **B, C** Exemplar TF assay results for (**B**) RcSxph and (**C**) NpSxph in the presence of STX, GTX5, C1/C2, dcGTC2/3, GTX2/3, or neoSTX at 0 nM (black) 19.5 nM (blue), 625 nM (cyan), 5000 nM (orange), and 20,000 nM (red) of each toxin. **D, E** Concentration dependence of ΔTm for (**D**) RcSxph and (**E**) NpSxph for STX (black circles), dcSTX (blue circles), GTX5 (green triangles), GTX2/3 (orange inverted triangles), dcGTX2/3 (inverted magenta triangles), C1/C2 (purple diamonds), neoSTX (dark red squares). Error bars are S.E.M. n = 2–8. Source data are provided as a Source Data file. Specific 'n' values are contained therein.

PSTs[17,23]. Together, our findings establish a molecular foundation for STX congener binding to Sxphs that should inform studies of 'toxin sponge'-based resistance mechanisms[18,24–26] and enable efforts to develop biologics to detect and neutralize STX and related PSTs.

# Results

## *Rc*Sxph and *Np*Sxph bind diverse STX congeners but not neosaxitoxins

To profile *Rc*Sxph[14,27] and *Np*Sxph[13] binding to different STX congeners, we used a TF assay[13,22] in which toxin binding causes a concentration-dependent change in the apparent melting temperature (Tm) of the target protein (Fig. 1A–E). We found that decarbamoylation at R1 (dcSTX) and alterations such as carbamate sulfation (GTX5) at R1, C11 sulfation at R3 (GTX2/3), combined C11 sulfation and decarbamoylation (dcGTX2/3), and combined carbamate and C11 sulfation (C1/C2) (Fig. 1A) shift the apparent melting temperature (ΔTm) (Table S1) of both Sxphs (Fig. 1B–E and Table S1). Among the various congeners, GTX5 behaved most like STX, having a similar ΔTm for both Sxphs (ΔTm = 3.7 ± 0.1 and 3.6 ± 0.2 °C, and 3.3 ± 0.1 and 3.2 ± 0.2 °C, for GTX5 and STX, and *Rc*Sxph and *Np*Sxph, respectively). By contrast, dcSTX, GTX2/3, dcGTX2/3, and C1/C2 exhibited reduced ΔTm changes, indicating that these congeners bind more weakly than STX (ΔTm = 3.1 ± 0.1, 2.7 ± 0.1, 2.3 ± 0.1, and 2.6 ± 0.1 °C, for *Rc*Sxph, respectively and ΔTm = 2.2 ± 0.1, 2.2 ± 0.1, 1.4 ± 0.1, and 1.5 ± 0.1 °C for *Np*Sxph, respectively), (Fig. 1B, C and Table S1). *Rc*Sxph and *Np*Sxph show similar ΔTm concentration dependences for this toxin panel (Fig. 1D, E). Importantly, these data establish that the Sxph TF assay can discriminate among the various modified forms of STX and serve as a facile means to assess Sxph:STX congener interactions.

Hydroxylation of the six membered guanidinium ring at N1 (the R2 site) yields neoSTX[2,5], a toxin that is a more effective Na$_v$ blocker[16] and more potent poison[1,5] than STX. To address whether *Rc*Sxph and *Np*Sxph bind N1 hydroxylated toxin congeners, we used the TF assay to profile neoSTX, dc-neoSTX, GTX6, and GTX1/4 (Figs. 1B–E and S1A–D, and Table S1). In contrast to the results obtained with STX congeners, neither neoSTX nor its derivatives affected ΔTm substantially (Fig. S1A–D and Table S1), indicating that N1 hydroxylation potently interferes with binding to *Rc*Sxph and *Np*Sxph. Although none of these toxins altered the *Np*Sxph ΔTm, neoSTX, dc-neoSTX, and GTX6 showed some minor ΔTm changes for *Rc*Sxph at the highest tested concentrations (>10 μM) (Fig. S1C). The *Rc*Sxph ΔTm rank order for dcSTX, GTX5 (B1), C1, and neoSTX (Table S1) agrees with radioligand binding competition studies of *R. catesbeiana* plasma[15] and reinforces prior conclusions that N1 (neoSTX) and C11 (C1) modifications have the largest impact on *Rc*Sxph binding[15].

To measure the affinity of STX and neoSTX congeners to *Rc*Sxph and *Np*Sxph, we modified our toxin binding FP assay[13,22] into a fluorescence polarization competition (FPc) assay[28] that measures the ability of unlabeled toxins to compete with a fluorescein-labeled STX derivative, F-STX[13] (Fig. 2A, B). Control experiments with STX yielded apparent affinities for *Rc*Sxph and *Np*Sxph (Kd = 8.9 ± 1.6 and 8.7 ± 1.7 nM, respectively) that correspond with direct FP measurements (Kd = 7.4 ± 2.6 and 0.5 ± 0.3 nM, respectively). Subsequent FPc profiling of STX congeners (Fig. 2A, B and Table S2) revealed trends that follow the TF results (Fig. 2C, D). As expected from its STX-like effects on ΔTm, GTX5 had an affinity similar to STX (Kd = 9.6 ± 0.8 and 8.9 ± 1.6 nM and 7.3 ± 1.8 and 8.7 ± 1.7 nM, for GTX5 and STX, and *Rc*Sxph and *Np*Sxph, respectively), whereas dcGTX2/3, the toxin having the smallest ΔTm, had the lowest affinity (180 ± 12 and 568 ± 68 nM for *Rc*Sxph and *Np*Sxph, respectively) (Table S2). neoSTX, dc-neoSTX, and GTX1/4 bound *Rc*Sxph and *Np*Sxph weakly, having affinities >5 μM (Fig. S1E, F and Table S2). In this panel, GTX6 was notable for displaying a weak but measurable affinity compared to other neoSTX congeners (1516 ± 222 and 1239 ± 578 nM for *Rc*Sxph and *Np*Sxph, respectively). However, this effect could be attributed to a ~0.5% contamination of

the strong binder, GTX5, in the commercial toxin sample used in this assay (Table S2). These results highlight the sensitivity of our assay and the ability of Sxphs to select between toxins in mixtures. The excellent agreement between ΔTm and the ΔΔG values (Fig. 2C, D) obtained by FPc aligns with TF-FP correlations observed in *Rc*Sxph STX binding pocket alanine scans[13] and demonstrates that ΔTm provides an accurate rank order assessment of STX congener affinity differences. Further, there is a strong correlation in STX congener binding between *Rc*Sxph and *Np*Sxph (Fig. 2E) that aligns with the similar STX binding pockets found in these two Sxphs[13].

## *Rc*Sxph and *Np*Sxph discriminate among STX congeners in a receptor binding assay

Testing for PST food contamination often relies on a radioligand RBA using tritiated STX ([³H]STX) and rat brain homogenate[17,29]. As Sxphs exhibit high thermostability, having Tms ~>50 °C[13], we wanted to examine whether purified Sxphs could be used in place of brain homogenates. RBAs using *Rc*Sxph (Figs. 2F and S2A–C) and *Np*Sxph (Figs. 2G and S2D–F) show that both proteins detect STX and its congeners consistent with their performances in the TF and FPc assays. The data reflect clear differences among the various toxins, including those from the neoSTX series (Table S3). Importantly, none of the toxins bind to *Rc*Sxph E540A, a mutant having ~2000-fold lower affinity for STX than *Rc*Sxph[13] (Fig. S2G). This result demonstrates that, as with the TF and FP assays[13], toxin detection relies on an Sxph STX binding pocket competent for toxin binding. RBA for *Rc*Sxph and *Np*Sxph and rat brain homogenate display comparable binding affinities, spanning 3–4 orders of magnitude (Kds ~1 nM–1 μM and ~3 nM-10 μM for rat brain homogenate and Sxphs, respectively) (Fig. S2H, I). Rat brain homogenates have a similar dynamic range for both the STX series (STX, dcSTX, GTX5, GTX2/3, dcGTX2/3, and C1/C2) and the neoSTX series (neoSTX, dc-neoSTX, GTX6, and GTX1/4) (Fig. S2H–K), whereas Sxphs are better at discriminating between these two toxin groups (Fig. S2A–F and Table S3). The RBA binding affinities and ΔΔG changes are well-correlated with those measured by FPc (Fig. 2H–I and Tables S2 and S3) providing further validation of the TF and FPc assay trends (Fig. 2C, D). Together, these data demonstrate that purified Sxphs can serve as viable replacements for rat brain homogenate in the RBA PST detection assay.

## Thermodynamics of toxin binding reveal critical moieties for Sxph interaction

To examine the thermodynamics of STX congener binding, we conducted ITC experiments to measure the binding of dcSTX, GTX2/3, dcGTX2/3, GTX5, and C1/C2 to *Np*Sxph directly (Table 1 and Fig. 3A–E). As expected from prior studies with STX[13–15,30], this STX congener set binds *Np*Sxph with a 1:1 stoichiometry (Table 1). GTX5 showed thermodynamic binding parameters similar to those of STX, in line with the comparable affinities of these two toxins, whereas the other toxins showed reduced Kds and concomitant ΔH and ΔS changes. The ΔΔG values determined by ITC follow those from the FPc assay, but show systematically tighter binding (Fig. 3F and Tables 1 and S2).

Comparison of the *Np*Sxph:STX congener binding free energy reveals interesting trends regarding the various STX modifications. Carbamate removal (dcSTX and dcGTX2/3) reduces binding by ~1 kcal mol⁻¹ consistent with this moiety interacting with key STX binding determinants Ile559, Phe562, and Pro728 within the *Np*Sxph binding pocket[13]. C11 sulfation is uniformly detrimental, causing ΔΔG changes of ~2–3 kcal mol⁻¹ for GTX2/3 vs. STX, dcGTX2/3 vs. dcSTX, and C1/C2 vs. GTX5. This modification makes close contacts with the binding pocket Gly798-Val799 backbone. By contrast, R1 carbamate sulfation (Fig. 1A) is favorable compared to STX (ΔΔG = −0.2 kcal mol⁻¹ for GTX5 vs. STX) but reduces binding when C11 is also sulfated (ΔΔG = 0.7 kcal mol⁻¹ for C1/C2 vs. GTX2/3). Hence, each modification evokes specific changes to the Sxph:toxin interaction. Some are straightforward to interpret, such as a reduction in binding enthalpy for dcSTX versus STX consistent with the

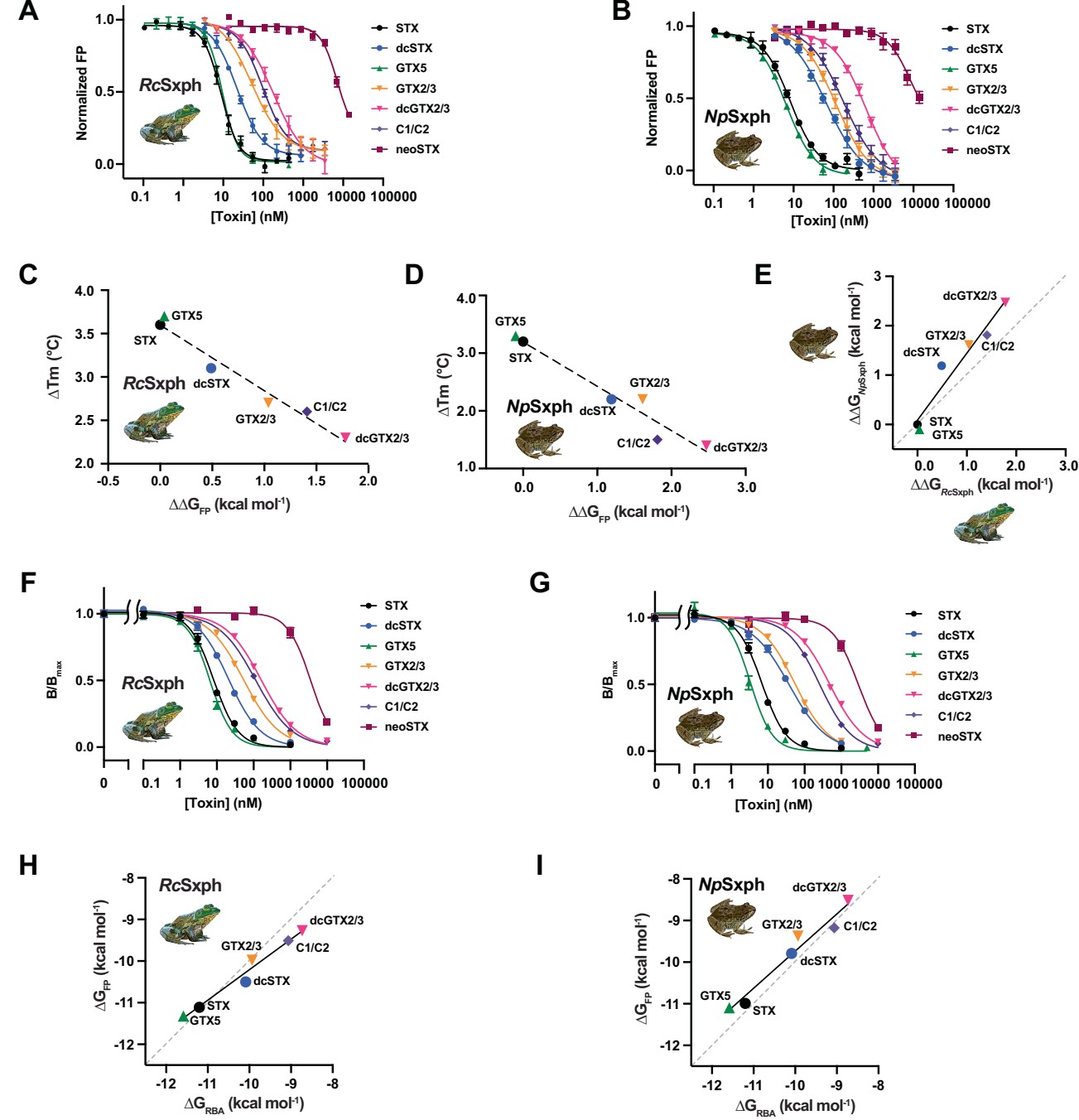

**Fig. 2 | FP and RBA studies reveal Sxph gonyautoxin binding affinities.**
**A, B** Exemplar competition FP assays for (**A**) *Rc*Sxph and (**B**) *Np*Sxph for STX (black circles), dcSTX (blue circles), GTX5 (green triangles), GTX2/3 (orange inverted triangles), dcGTX2/3 (inverted magenta triangles), C1/C2 (purple diamonds), and neoSTX (dark red squares). **C, D** Comparisons of ΔTm and ΔΔG values for (**C**) *Rc*Sxph and (**D**) *Np*Sxph. (line $y = -0.7564x + 3.6$ $R^2 = 0.9668$ and $y = -0.7679x + 3.193$ $R^2 = 0.9485$, respectively). **E** Comparison of ΔΔG relative to STX for binding of the indicated toxins to *Np*Sxph and *Rc*Sxph (line, $y = 1.356x + 0.08794$ $R^2 = 0.9417$). Grey dashed line shows $x = y$. **F, G** Exemplar RBA

curves for (**F**) *Rc*Sxph and (**B**) *Np*Sxph for STX (black circles), dcSTX (blue circles), GTX5 (green triangles), GTX2/3 (orange inverted triangles), dcGTX2/3 (inverted magenta triangles), C1/C2 (purple diamonds), and neoSTX (dark red squares) where B/B$_{max}$ represents the bound [$^3$H]STX in the sample/maximum binding of [$^3$H]STX in the absence of competing unlabeled STX. **H, I** Comparison of binding free energies measured by FP and RBA assays for (**H**) *Rc*Sxph and (**I**) *Np*Sxph. (line $y = 0.7331x - 2.878$ $R^2 = 0.9755$ and $y = 0.89314x - 0.8046$ $R^2 = 0.9604$, respectively). Grey dashed line shows $x = y$. Error bars are S.E.M. $n = 3$–8. Source data are provided as a Source Data file. Specific '$n$' values are contained therein.

fewer dcSTX contacts to the Sxph binding site, whereas others, such as the C11 sulfated toxins, show more complex enthalpy-entropy compensation effects suggesting that changes in water solvation of the toxin and binding pocket are important[31,32]. The *Np*Sxph TF, FPc, RBA, and ITC results are in excellent agreement, and together provide a versatile assay suite for characterizing Sxph:toxin interactions[22].

## *Np*Sxph:STX congener structures reveal similar interactions and a common water network

To visualize Sxph:STX congener interactions, we co-crystallized and determined the X-ray crystal structures of *Np*Sxph:dcSTX, *Np*Sxph:GTX2, *Np*Sxph:dcGTX2, *Np*Sxph:GTX5, and *Np*Sxph:C1 at resolutions of 1.90 Å, 1.95 Å, 1.80 Å, 1.90 Å, and 1.90 Å, respectively,

**Table 1 | *Np*Sxph:Toxin thermodynamic binding parameters**

| | | N (sites) | Kd (nM) | ΔH (kcal mol⁻¹) | ΔS (cal mol⁻¹ K⁻¹) | ΔG (kcal mol⁻¹) | ΔΔG (kcal mol⁻¹) | n |
|---|---|---|---|---|---|---|---|---|
| *Np*Sxph | STX[a] | 0.92 ± 0.02 | 2.5 ± 0.1 | −18.7 ± 0.2 | −23.2 ± 0.8 | −11.8 ± 0.1 | – | 2 |
| | dcSTX | 0.81 ± 0.01 | 11.3 ± 4.0 | −17.8 ± 0.4 | −23.3 ± 2.0 | −10.9 ± 0.2 | 0.9 | 2 |
| | GTX2/3 | 1.11 ± 0.15 | 78.2 ± 18.1 | −11.9 ± 1.3 | −7.3 ± 4.1 | −9.7 ± 0.1 | 2.1 | 2 |
| | dcGTX2/3 | 1.00 ± 0.13 | 295 ± 74 | −12.4 ± 0.0 | −11.7 ± 0.4 | −8.9 ± 0.1 | 2.9 | 2 |
| | GTX5 | 0.90 ± 0.06 | 1.8 ± 0.8 | −18.0 ± 0.8 | −20.2 ± 3.5 | −12.0 ± 0.3 | −0.2 | 2 |
| | C1/C2 | 0.86 ± 0.10 | 274 ± 71 | −14.3 ± 2.2 | −17.7 ± 8.0 | −9.0 ± 0.2 | 2.8 | 2 |

*Np*Sxph: Toxin Thermodynamic Parameters. 'N' is the number of binding sites. Kd denotes the dissociation constant. ΔΔG = ΔG$_{Toxin}$–ΔG$_{STX}$. 'n' is the number of observations.
[a]data from ref. 13 Errors are S.D.

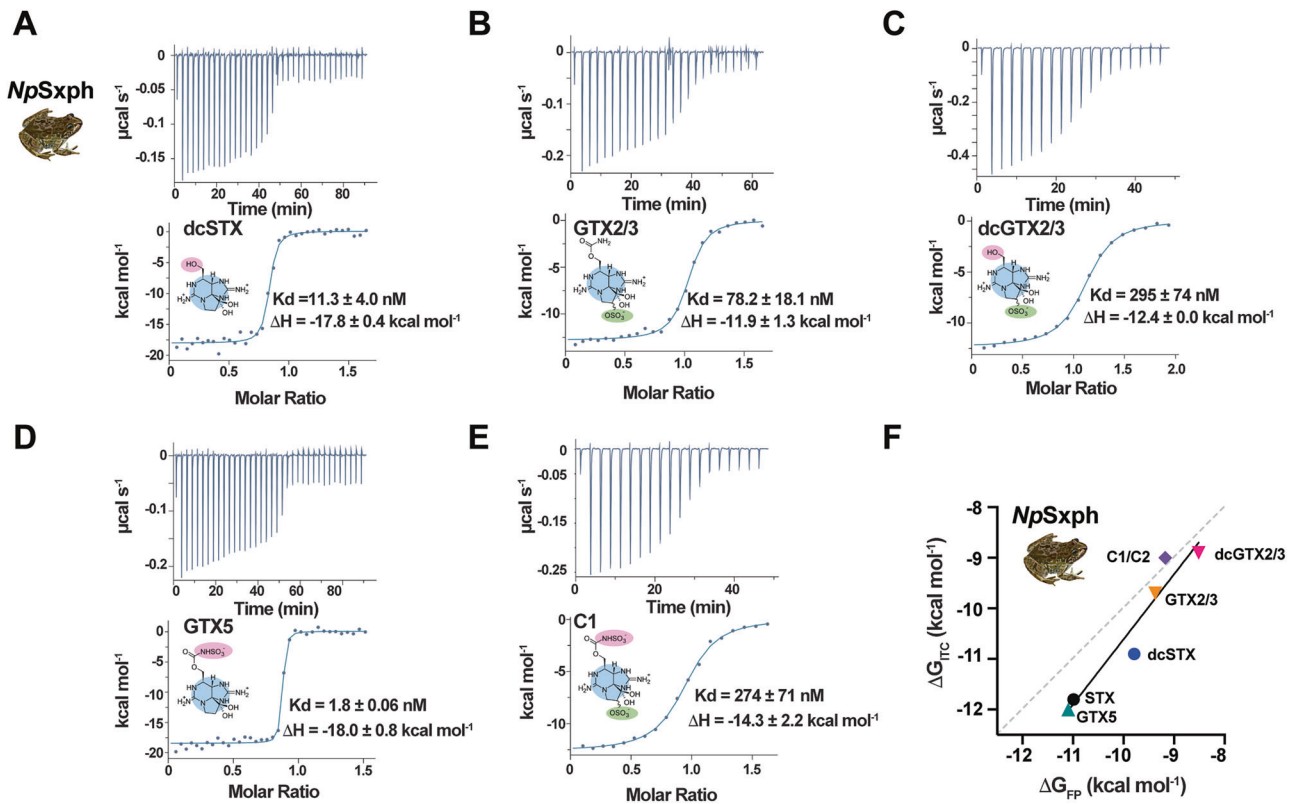

**Fig. 3 | ITC studies reveal *Np*Sxph STX congener binding thermodynamic properties. A–E** Exemplar ITC isotherms for the titration of (**A**) 100 µM dcSTX into 11.4 µM *Np*Sxph (**B**) 100 µM GTX2/3 into 11.3 µM *Np*Sxph (**C**) 200 µM dcGTX2/3 into 22.4 µM *Np*Sxph (**D**) 100 µM GTX5 into 12.4 µM *Np*Sxph and **E** 100 µM C1/C2 into 11.9 µM *Np*Sxph. Kd and ΔH values are indicated. Toxin structures are shown. **F** Comparison of ΔG$_{ITC}$ and ΔG$_{FP}$ for *Np*Sxph binding to each of the indicated toxins (line y = 1.287x + 2.263, R² = 0.9287). Grey dashed line shows x = y. Source data are provided as a Source Data file.

using molecular replacement (Table S4). The maps show excellent quality densities that correspond to the unique shape of each bound toxin (Fig. 4A–E) and show that all STX congeners bind *Np*Sxph with a 1:1 stoichiometry matching the functional studies (Table 1). GTX2/3, dcGTX2/3, and C1/C2 are epimeric mixtures at the C11 sulfated position[1,2,33]. Because it was not possible to discern the bound epimer from the density, we assigned the density to the most prevalent epimer in the sample (GTX2, dcGTX2, and C1) (Fig. 4B, D, and E).

Comparisons of the *Np*Sxph:STX congener structures with the *Np*Sxph:STX complex (PDB: 8D6M)[13] show that there are no large-scale changes in the Sxph core (N- and C-lobes) and thyroglobulin domains (Thy1-1 and Thy1-2) (root mean square deviation of Cα positions (RMSD$_{Cα}$) = 0.229 Å, 0.259 Å, 0.223 Å, 0.235 Å, and 0.239 Å for dcSTX, GTX2, dcGTX2, GTX5, and C1, respectively) (Fig. 4F). Superposition of STX and STX congeners highlights that all have a similar binding pose

involving identical interactions of the tricyclic bis-guanidinium core with *Np*Sxph Glu541, Asp786, Asp795, and Tyr796 that follow the STX molecular recognition fingerprint[13] (Figs. 4G and S5A–F). In general, there are only minor changes in the position of the toxin among the structures. The most variation occurs at the carbamate (Cb) (Fig. 4H), with the largest shifts observed for the sulfocarbamate toxins GTX5 and C1 (- 0.9 Å and -1.8 Å, respectively).

Despite the variety of STX core modifications, the *Np*Sxph STX binding pocket residues display minimal structural movement between the apo[13] and STX congener bound forms (Movie S1). In all structures, Asp786 coordinates the five-membered ring as seen in the *Np*Sxph:STX complex[13] (Fig. S4A–F) and is repositioned from the outward-facing rotamer of the apo structure (Fig. S4G)[13]. In the *Np*Sxph:GTX5 and *Np*Sxph:C1 complexes, Arg719 moves by -5 Å towards the Cb sulfate (Figs. 4G, S4C, and F and Movie S1) and together

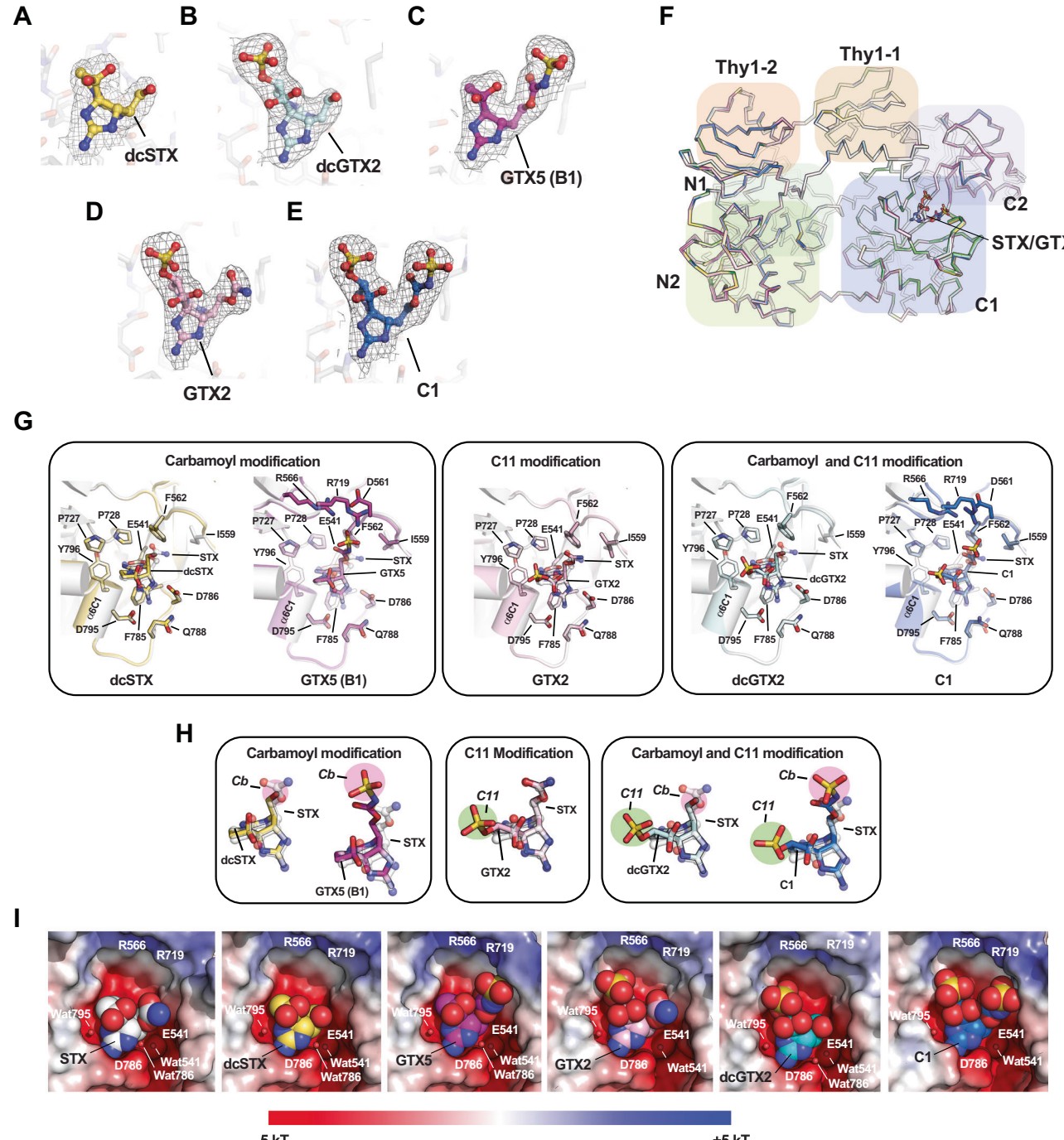

**Fig. 4 | Structures of *Np*Sxph:STX congeners complexes. A–E** Exemplar 2Fo-Fc (1σ) densities for (**A**) dcSTX (yellow), (**B**) dcGTX2 (cyan), (**C**) GTX5 (B1) (magenta), (**D**) GTX2 (light pink), and (**E**) C1 (marine). **F** Superposition of *Np*Sxph:STX (PDB:8D6M)[13] (white). *Np*Sxph:dcSTX (yellow). *Np*Sxph:GTX2 (light pink), *Np*Sxph:dcGTX2 (cyan), *Np*Sxph:GTX5 (magenta), and *Np*Sxph:C1 (marine). N1 (green), N2 (yellow green), C1 (blue), C2 (purple), Thy1–1 (tan), and Thy1-2 (orange) domains are indicated. STX and congeners are shown as sticks. **G** Comparisons of the toxin binding pocket from the *Np*Sxph:STX complex (PDB:8D6M)[13] with *Np*Sxph:STX congener structures. Structures are grouped by STX modification type: Carbamoyl modification, dcSTX (yellow) and GTX5(B1) (magenta); Carbamoyl and C11 modification, dcGTX2 (cyan) and C1 (marine); and C11 modification, GTX2 (light pink). **H** Comparison of the STX congeners from (**F**) with STX from the *Np*Sxph:STX complex (PDB:8D6M)[13]. STX congener colors are as in (**F**). Modification sites R1 (carbamoyl) (magenta) and R3 (C11) (green) are highlighted. **I** Electrostatic surface potentials for *Np*Sxph:STX complex (PDB:8D6M)[13] and the indicated *Np*Sxph:STX congener complexes. STX and STX congeners are shown in space filling.

with Arg566 forms an electrostatically positive ridge near the Cb sulfate (Fig. 4I). Both residues are conserved among Sxph sequences[13].

The high resolution structures also revealed a set of water-mediated networks that contribute to *Np*Sxph:toxin interactions that we identify based on the residue with which they interact. Two water molecules, Wat541 and Wat795, occupy the same positions in the *Np*Sxph:STX complex[13] and all *Np*Sxph:STX congener structures and form interactions with key hot-spot residues[13] (Figs. S3A–E, S4A–G, and S5A–F). Wat541 is deeply buried in the binding pocket, and bridges the Glu541 sidechain, the residue that contributes the most binding energy to STX core recognition[13], with the backbone carbonyl of Glu784 (Fig. S4H). Notably, this water bridge is absent in the apo structure[13]

(Fig. S4G). Instead, there is an axial water (Wat541a) on the opposite side of Glu541 (Fig. S4I). The second key water molecule, Wat795, bridges the sidechain and backbone oxygens of Asp795 with STX OH-14 and STX N9 (Fig. S4J) and is also lacking in the apo structure (Fig. S4G). The Asp795 site contributes nearly as much binding energy to toxin binding as the Glu541 site (~ 4.5 kcal mol$^{-1}$)[13]. Thus, the two most important residues for binding the STX core, Glu541 and Asp795, rely on water-mediated interactions. All structures, except those with C1 and GTX2, revealed a third water molecule, Wat786, coordinated with the Asp786 sidechain and the N7 position on the five-membered toxin ring (Figs. S3A–C, S4J, and S5). This Asp786 residue is one of the few residues that move between the apo and bound forms[13,14] but contributes little to toxin binding energetics[13].

To examine the roles of these water molecules further, we conducted a set of MD simulations totaling 5 µs for $Np$Sxph:STX and each of the $Np$Sxph:STX congener complexes, as well as models of complexes with the C11 epimers GTX3, dcGTX3, and C2. The STX core (Fig. S6A, B) and STX binding pocket (Fig. S6C, D) show remarkably low mobility leading to very stable toxin:protein contacts that persist throughout the simulations, regardless of toxin type (Fig. S6E). The similar behavior of the GTX2/3, dcGTX2/3, and C1/C2 epimers (Fig. S6A–C) indicates that there is little discrimination between these forms as underscored by their similar contacts in all trajectories (Fig. S6E). The stability of the Sxph:toxin interactions extends to the key water molecules Wat541 and Wat795, that coordinate Glu541 and Asp795 and their interactions with the toxin (Fig. S6E, F). Notably, Wat795 shows lower occupancy in the C11 sulfated toxin complexes (Fig. S6F), suggesting that part of the affinity loss caused by the sulfate modification derives from its effect on water occupancy at this site. This Wat795 behavior agrees with the observation that C11 sulfated congeners show a more favorable binding entropy (Table 1) and supports the idea that toxin and binding pocket solvation changes contribute to toxin affinity. Hence, the mobility of this key water molecule is influenced by the STX congener identity. In line with the lesser energetic importance of the Asp786 position[13], its bound water, Wat786, exhibits low occupancy (Fig. S6F). Further, the absence of Wat786 is frequently correlated with the Asp786 movement away from the toxin and hydration of the coordination site by bulk water. Taken together, the X-ray data and MD simulations highlight the structurally rigid, pre-organized nature of the Sxph binding pocket and identify a shared set of protein-toxin and water-mediated contacts that engage diverse STX congeners having affinities that vary by ~two orders of magnitude.

## Sxphs rescue Na$_V$s from STX congener block

$Rc$Sxph is a 'toxin sponge' that can reverse STX inhibition of Na$_V$s by competing for STX[13,18]. As this ability is linked to Sxph affinity for STX[13] and $Np$Sxph binds STX better than $Rc$Sxph by ~15 fold[13], we wanted to test whether $Np$Sxph could act as a more effective toxin sponge. To this end, we used a previously established TEVC assay[13,18] in which 100 nM STX yields 90% block (IC$_{90}$) of the Golden poison frog $Phyllobates$ $terribilis$ skeletal muscle channel Na$_V$1.4 ($Pt$Na$_V$1.4) expressed in $Xenopus$ $laevis$ oocytes to measure the effect of Sxph addition (Figs. 5A, B and S7A, B). Consistent with their shared low nanomolar STX affinity[13], Sxph titration showed that 2:1 Sxph:toxin ratios were sufficient for both $Np$Sxph and $Rc$Sxph to reverse STX block completely (Fig. 5A–C), agreeing with prior $Rc$Sxph studies[18]. Notably, the Sxph ratio required for 50% rescue of the blocked current was better for $Np$Sxph (Effective Rescue Ratio 50, ERR$_{50}$ = 0.5 ± 0.1 for $Np$Sxph and 0.7 ± 0.2 for $Rc$Sxph) (Figs. 5A–C and S7A, B), in agreement with its higher STX affinity[13].

As $Np$Sxph and $Rc$Sxph[13,18] both act as effective anti-STX toxin sponges (Fig. 5A–C) and also have strong binding affinity for some STX congeners (Tables 1 and S1–S3), we next asked whether these proteins could act more broadly to rescue Na$_V$s from different STX family members. We first measured the $Pt$Na$_V$1.4 response to dcSTX, GTX2/3, and GTX5 (Fig. 5D, E) using TEVC. All of these toxins were weaker blockers relative to STX (IC$_{50}$ = 12.6 ± 1.4 nM[18]) having a rank order of STX < GTX2/3 < dcSTX << GTX5 (IC$_{50}$ = 27.5 ± 2.1, 144.4 ± 29.7, and 2290 ± 563 nM, respectively) (Fig. 5D, E and Table S5) similar to mammalian Na$_V$1.4[33–36]. We then tested whether a 3:1 $Np$Sxph:toxin ratio could neutralize GTX2/3 and dcSTX block of $Pt$Na$_V$1.4 at their respective IC$_{90}$s (200 nM and 800 nM, respectively). GTX5 was not tested because its weak IC$_{90}$ (>10 µM) required prohibitively large amounts of $Np$Sxph and toxin in this assay. These experiments revealed that $Np$Sxph reverses channel inhibition by both GTX2/3 (Fig. 5F) and dcSTX (Fig. 5G). Further, in line with its lower affinity for GTX2/3 relative to STX, $Np$Sxph had an increased ERR$_{50}$ for this toxin (ERR$_{50}$ = 2.1 ± 0.7 and 0.5 ± 0.1, respectively) (Figs. 5E and S7C), providing further support that rescue effectiveness depends on Sxph toxin affinity[13].

Comparison of rescue kinetics revealed unexpected differences between dcSTX and GTX2/3. A 3:1 $Np$Sxph:toxin ratio caused rapid reversal of dcSTX block and complete current recovery within one minute (102.6 ± 1.2%, $n$ = 6) (Figs. 5G and S7D). By contrast, reversal of GTX2/3 block using the same Sxph:toxin ratio was slower and biphasic showing an initial fast recovery phase yielding 87.4 ± 6.5% ($n$ = 6) current recovery after 5 min (Fig. S7C and E) followed by a slower recovery phase that reached 91.8 ± 2.8% ($n$ = 2) at 30 min (Figs. 5F and S7E). To test if the two phases resulted from contaminants in the naturally isolated GTX2/3 sample (GTX2/3$_{nat}$), we repeated the experiment using pure, synthetic GTX2/3 (GTX2/3$_{syn}$). These experiments showed that the 3:1 $Np$Sxph:GTX2/3$_{syn}$ ratio had identical reversal behavior to GTX2/3$_{nat}$, displaying a biphasic response reaching 87.3 ± 3.3% ($n$ = 5) current rescue at 5 min and a slower rescue phase that reached 94.4 ± 3.3% ($n$ = 5) at 30 min (Figs. S7F, G). Having eliminated toxin contaminants as the source of the two phases, we next tested whether $Rc$Sxph would behave comparably given that it has a similar GTX2/3 binding affinity to $Np$Sxph (Tables S2 and S3). Matching the effect of $Np$Sxph, a 3:1 $Rc$Sxph:GTX2/3$_{syn}$ ratio was sufficient for complete rescue of channel block (Fig. S7H, I). However, $Rc$Sxph showed a faster current recovery during the first phase (97.2 ± 4.5% ($n$ = 4) after 5 min) and complete rescue after 10 minutes (100.6 ± 2.8%, $n$ = 4) (Fig. S7H). Structural comparison suggests that the GTX2/3 selectivity differences may originate from more electropositive surface near the C11 sulfate found in the $Rc$Sxph STX binding pocket compared to $Np$Sxph (Fig. S7J). Together, these results demonstrate that Sxphs rescue Na$_V$ from STX congener block and uncover functional differences between Sxphs having similar STX binding sites[13,14] (Tables S1–S3, and S5).

## Sxph:Toxin binding neutralizes human Na$_V$ poisoning by diverse STX congeners

To overcome the sample limitations posed by the TEVC assay and enable investigation of a wider range of toxins, we established a planar whole cell patch-clamp assay using mammalian cells stably expressing $Homo$ $sapiens$ Na$_V$1.4 ($Hs$Na$_V$1.4) that required ~10,000-fold lower test sample volumes than the TEVC assay. Profiling the IC$_{50}$ values of STX and five congeners against $Hs$Na$_V$1.4 (Fig. 6A, B and Table S5) showed that low nanomolar concentrations of STX, GTX2/3, dcSTX, and dcGTX2/3 inhibited $Hs$Na$_V$1.4 (IC$_{50}$ = 3.0 ± 1.6, 6.8 ± 1.1, 14.7 ± 8.7, and 31.8 ± 10.6 nM respectively) consistent with prior measurements for $Hs$Na$_V$1.4[6]. These measurements also revealed modest changes in potency for decarbamoylation and C11 sulfation that match prior studies[3,34]. By contrast, carbamate sulfation on GTX5 and C1/C2 reduced $Hs$Na$_V$1.4 block by ~2 orders of magnitude relative to STX (IC$_{50}$ = 335.3 ± 55.4 and 151.5 ± 37 nM, respectively) (Fig. 6A, B and Table S5) consistent with previous reports[3,34].

We then assessed the ability of $Np$Sxph to prevent $Hs$Na$_V$1.4 block by STX, dcSTX, GTX2/3, dcGTX2/3, GTX5, and C1/C2 at their respective IC$_{90}$ values (Table S5). Comparing the $Hs$Na$_V$1.4 sodium currents in the

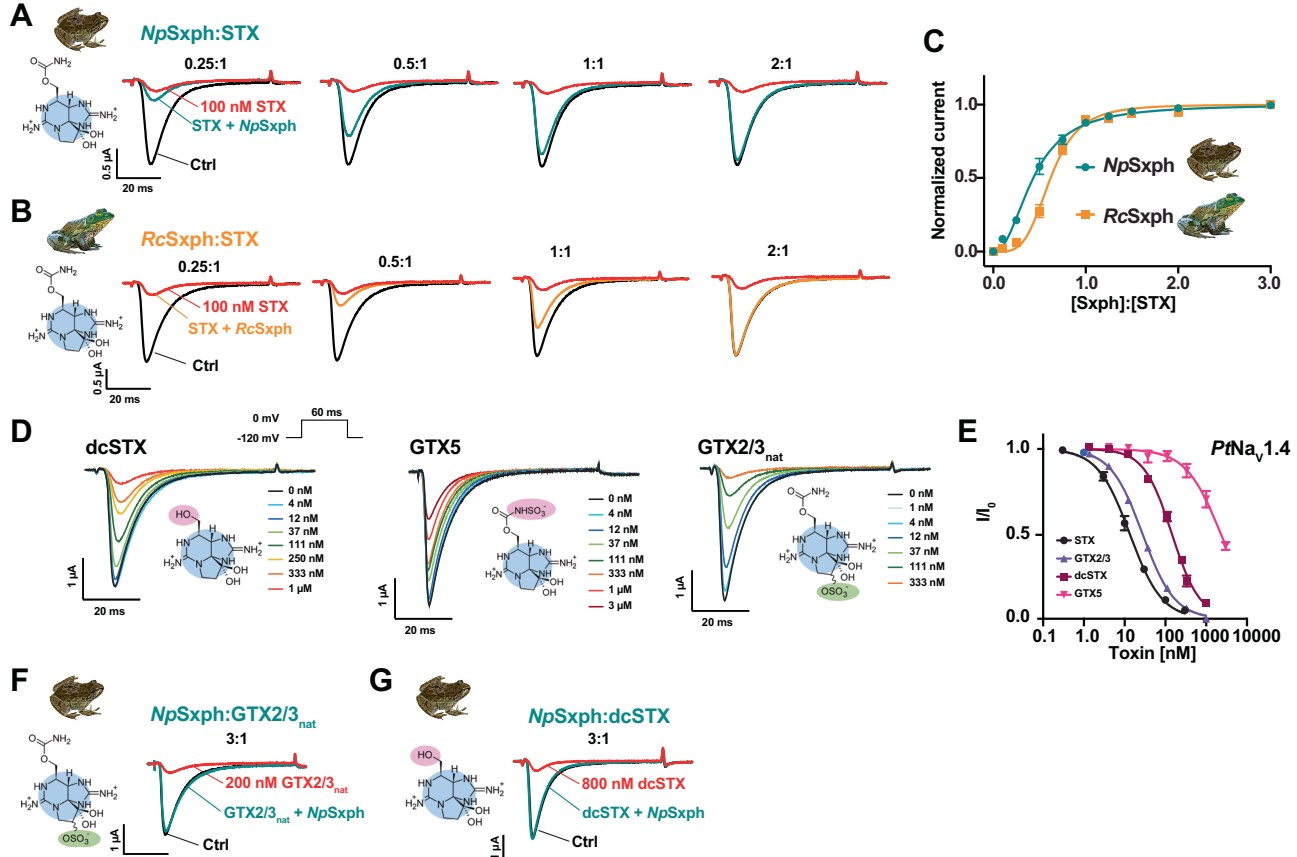

**Fig. 5 | NpSxph rescues PtNaV1.4 block by diverse STX congeners. A, B** Exemplar TEVC recordings of PtNaV1.4 before (Ctrl, black) and after (red) application of 100 nM STX, and after application of Sxph in the presence of 100 nM STX at the indicated [Sxph]:[STX] molar ratios for (**A**) NpSxph (teal) and (**B**) RcSxph (orange). **C** [NpSxph]:[STX] (teal) and [RcSxph]:[STX] (orange) dose-response curves in the presence of 100 nM STX. **D** Exemplar TEVC recordings of PtNaV1.4 expressed in X. laevis oocytes in the presence of the indicated STX congener (dcSTX, GTX5, and

GTX2/3) concentrations. Inset shows the voltage pulse protocol. **E** Toxin response curves for STX (black circles, from Abderemane-Ali et al., 2021), GTX2/3 (purple triangles), dcSTX (maroon squares), and GTX5 (magenta inverted triangles). **F, G** Exemplar PtNaV1.4 responses to application of (**F**) 200 nM GTX2/3 (red) (30 min post application), **G** 800 nM dcSTX (red) (one minute post application), and 3:1 [NpSxph]:[toxin] application (teal). For C and E, data represents mean with S.E.M. from 6–8 oocytes. Source data are provided as a Source Data file.

presence of a 5:1 NpSxph:toxin mixture against those elicited in the absence and presence of each toxin at its IC$_{90}$ concentration revealed that NpSxph neutralized all tested STX congeners (Fig. 6C, D). As with the TEVC rescue experiments[13] (Fig. 5C), the degree of neutralization followed toxin binding affinity trends. We observed nearly complete neutralization of STX, GTX5, C1/C2, and dcSTX (93.7 ± 6.2, $n = 5$; 99.2 ± 1.4, $n = 4$; 90.4 ± 5.8, $n = 6$; and 88.3 ± 5.5%, $n = 7$; respectively) (Fig. 6C, D) and partial neutralization of GTX2/3 and dcGTX2/3 (53.2 ± 9.5, $n = 7$; and 59.9 ± 7.0%, $n = 7$; respectively) (Fig. 6E). The lower fractional recovery for the C11 monosulfated toxins in the planar patch-clamp recordings highlights that rescue is governed by the relative affinities of a given toxin for the target channel and Sxph. Together with the TEVC studies (Fig. 5F, G), these data demonstrate the capacity of Sxphs to act as broad-spectrum 'toxin sponges' that can neutralize multiple, naturally-occurring STX congeners and reverse their inhibition of NaVs.

## Divergent STX congener affinities highlight Sxphs and NaVs toxin binding factors

Structural and binding studies have uncovered a remarkable convergence in the way Sxphs and NaVs recognize the STX core[13,14]. Our studies here reveal that despite this common recognition framework, there are clear differences in how R1, R2, and R3 toxin modifications (Fig. 1A) affect interaction with each target. The sulfocarbamate (GTX5) has minimal effect on Sxph binding (Tables S2 and S3) but is

detrimental to NaV1.4 binding (>3 kcal mol$^{-1}$) (Table S3). Conversely, C11 sulfation (GTX2/3) has modest effects on channel interactions but destabilizes binding to Sxphs by ~1 kcal mol$^{-1}$. Whereas N1 hydroxylation (neoSTX) increases toxin affinity (Table S3) and ability to block the channel[16], this modification severely compromises RcSxph and NpSxph binding (>3.7 kcal mol$^{-1}$) (Tables S2 and S3). There are no NaVs:STX congener structures. Thus, we used the NpSxph:C1 structure, having a toxin bearing sulfation at both the carbamate and C11 to compare the toxin pose with that found in the human NaV1.7:STX complex[11] (Fig. 7A, B) to gain insight into the STX congener binding differences between the two targets. The position of the DI and DII residues that engage the STX core sets a lateral toxin orientation across the NaV pore ring that leads to sulfocarbamate clashes with the P1 and P2 helices and NaV selectivity filter of the DIII pore module and exposure of the C11 sulfate to the extracellular solvent facing side of the pore above the DIV the selectivity filter (Fig. 7A, B). The putative sulfocarbamate-DIII clash and accommodation of the C11 sulfate are consistent with the effects of these modifications on NaV (Tables S3 and S5), common effects of DIII sequence variations on STX and GTX3 block[37], and energetic coupling between carbamate modification and DIII mutants[36]. By contrast, the Sxph residues that coordinate the STX core sit at the bottom of a deep pocket (Fig. 7C, D). This positioning sets a toxin orientation that exposes both the carbamoyl and C11 sulfate groups to the solvent-exposed opening of the binding site and explains why both toxin forms bind Sxphs. Notably, N1 hydroxylation appears

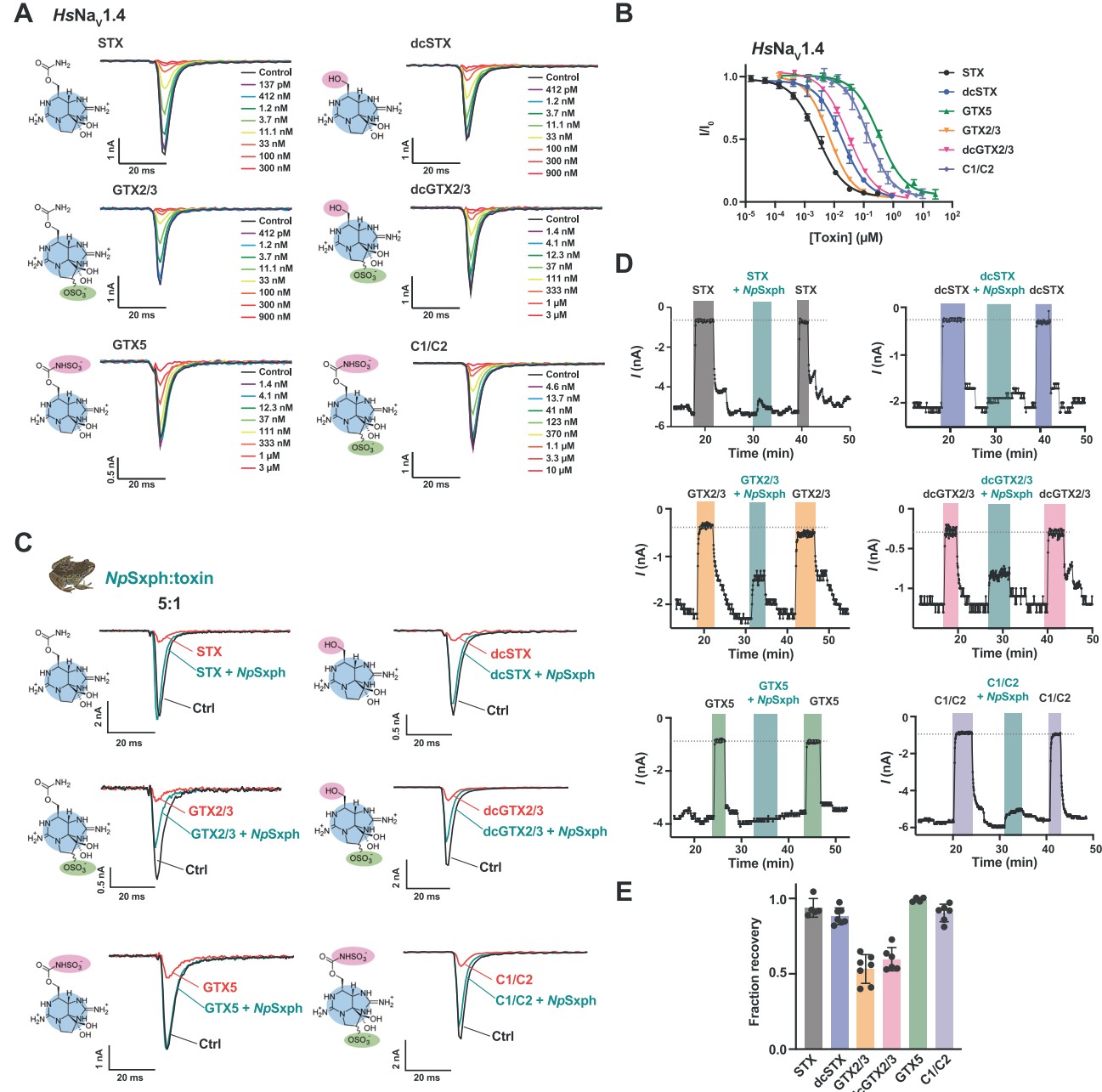

**Fig. 6 | *Np*Sxph rescues *Hs*Na_V1.4 from block by diverse STX congeners.**
**A** Exemplar whole-cell patch clamp recordings of *Hs*Na_V1.4 expressed in CHO cells in the presence of the indicated STX congener (STX, dcSTX, GTX2/3, dcGTX2/3, GTX5, and C1/C2) concentrations. **B** Toxin dose-response curves for STX (black circles), dcSTX (blue circles), GTX5 (green triangles), GTX2/3 (orange inverted triangles), dcGTX2/3 (magenta inverted triangles), and C1/C2 (purple diamonds). **C** Exemplar whole-cell patch clamp *Hs*Na_V1.4 responses in the absence (Ctrl, black) and presence (red) of 50 nM STX, 200 nM dcSTX, 60 nM GTX2/3, 300 nM dcGTX2/3, 1 μM C1/C2, and 2.7 μM GTX5, and then after 5:1 [*Np*Sxph]:[Toxin] application (teal). **D** Exemplar whole-cell patch clamp time course showing *Hs*Na_V1.4 peak

currents from (**C**) after application of STX congener (50 nM STX (grey), 200 nM dcSTX (blue), 60 nM GTX2/3 (orange), 300 nM dcGTX2/3 (pink), 1 μM C1/C2 (purple), and 2.7 μM GTX5 (green)), followed by toxin wash-out, then application of 5:1 [*Np*Sxph]:[Toxin] mixture (teal). Following the [*Np*Sxph]:[Toxin] wash-out, toxin at the desired concentration (50 nM STX (grey), 200 nM dcSTX (blue), 60 nM GTX2/3 (orange), 300 nM dcGTX2/3 (pink), 1 μM C1/C2 (purple), and 2.7 μM GTX5 (green)) was applied. **E** Fraction recovery for 5:1 *Np*Sxph:toxin. Colors as in (**D**). For B and E, error bars are S.E.M. *n* = 2–9. Source data are provided as a Source Data file. Specific '*n*' values are contained therein.

to create a clash with the key binding position Glu541[13] in the deepest part of the Sxph binding site (Figs. 7C, D and S8), offering a rationale for why neoSTX and its congeners are poor *Rc*Sxph and *Np*Sxph binders (Tables 1 and S1–S3). Together, these comparisons show that the orientation of the STX core components with respect to other binding target elements has a strong influence on how STX targets interact with various STX core modifications and identify key structural features that differentiate how Na_Vs and Sxphs bind STX congeners.

## Discussion

Sxphs[13,14] and Na_Vs[11,12,38] bind STX with high affinity by means of a rigid binding site that uses a convergent molecular recognition strategy to engage the STX bis-guanidinium core[13,14] even though each presents the shared STX binding motif on different scaffolds. How STX congener modifications affect binding interactions with these two classes of PST binding proteins and whether Sxph[13,14] and Na_V[11,12,38] high affinity STX sites can be reshaped to accommodate the various STX congener

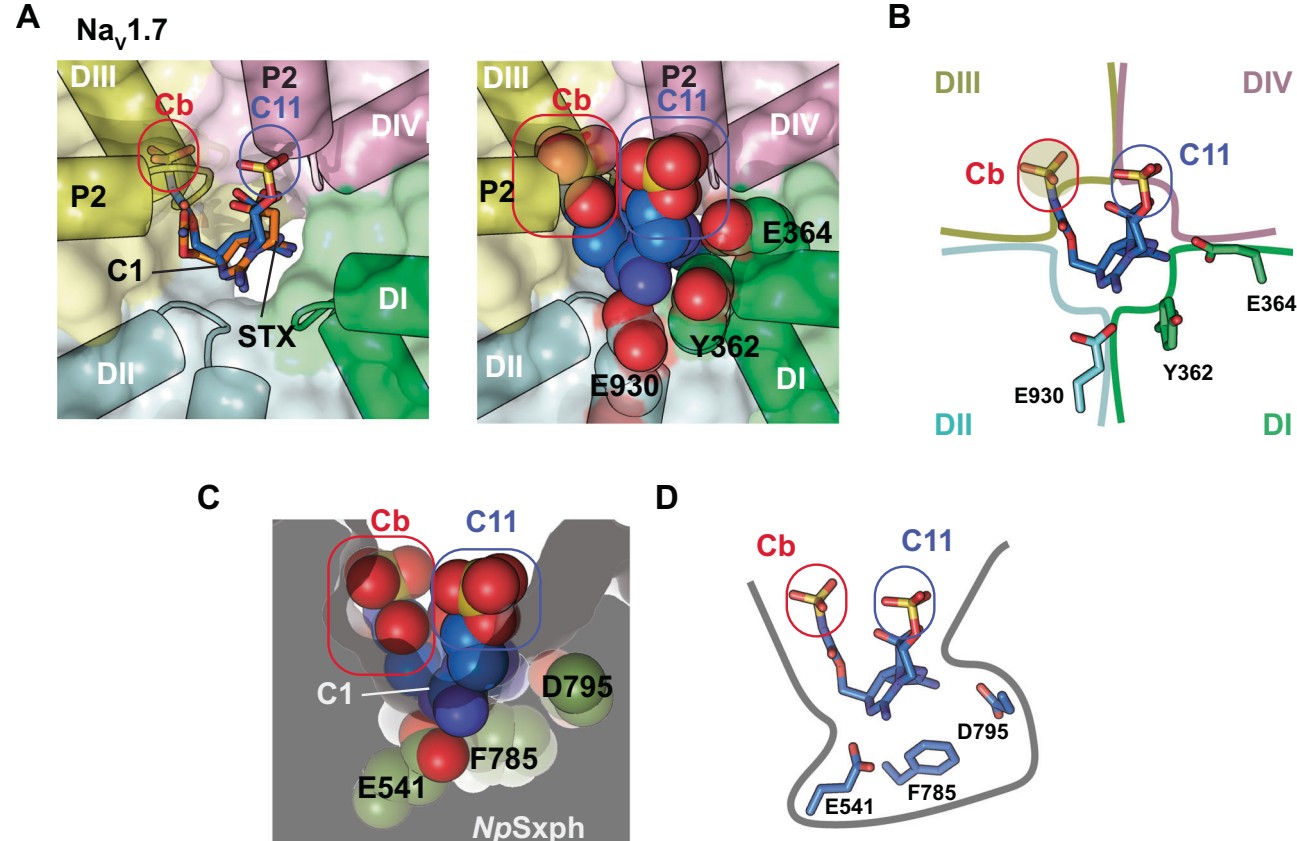

**Fig. 7 | Structural comparison of STX binding site in Na$_V$1.7 and *Np*Sxph. A** Top view of the STX binding site in Na$_V$1.7 structure in complex with STX (PDB:6J8G)[11]. **Left**, Comparison of STX (orange) with C1 (blue) shown in stick models. Modification sites for C1 are highlighted, sulfate on the carbamate (Cb, red) and sulfate on the C11 position (C11, blue). **Right**, C1 shown in space filling. **B** Schematic of C1 model from (**A**). Clash site in DIII is shaded. **C** Toxin binding site in *Np*Sxph:C1 complex (PDB:8V67). C1 shown in space filling. **D** Schematic of *Np*Sxph:C1 complex from (**C**).

modifications have been unclear. Our data show that the most prevalent STX congener alterations (decarbamoylation and carbamate sulfation at R1, N1 hydroxylation at R2, and C11 sulfation at R3) (Fig. 1A) have profound effects on *Rc*Sxph and *Np*Sxph toxin affinities that range over of ~3 orders of magnitude (Tables 1 and S1–S3). Strikingly, despite these large affinity differences, our structural analysis shows that STX congeners bearing various combinations of decarbamoylation, carbamate sulfation, and C11 sulfation bind with poses matching STX (Figs. 4G, H and S6A, B). Further, similar to STX[13], dcSTX, GTX2/3, dcGTX2/3, GTX5, and C1/C2 binding induces minimal conformational change in the Sxph toxin binding pocket, apart from the movement of an aspartate (*Np*Sxph Asp786) to coordinate the five-membered guanidinium ring of the toxin (Fig. S4 and Movie S1). The strongest binding perturbations are caused by N1 hydroxylation in the neoSTX series (Tables S1–S3) that creates an apparent clash in a deeply buried part of the binding pocket (Fig. S8). Thus, the 'lock and key' mode of toxin binding used by Sxph strongly influences toxin preferences.

Our structural studies (Figs. S3–S5) and MD simulations (Fig. S6) identified a set of common water molecules that are crucial to toxin binding. These ordered waters, absent from the apo-*Np*Sxph structure[13] (Fig. S4G), buttress toxin interactions by stabilizing the residue that interacts with the N1 nitrogen of the six membered guanidinium ring, *Np*Sxph Glu541, and by bridging *Np*Sxph Asp795 and the toxin hydrated ketone (Fig. S4A–J). Notably, these conserved Sxph residues are most critical for STX binding affinity[13], each contributing >4 kcal mol$^{-1}$ binding energy. Hence, maintaining 'lock and key' binding comes at the cost of reduced affinity for decarbamoylated, C11 sulfated, and N1 hydroxylated congeners (Tables 1 and S1–S3) as each of these modifications affects interactions with various elements

of the Sxph pocket. These affinity differences arise from loss of carbamate contacts[13] (dcSTX and dcGTX2/3), destabilization of the Asp795 water-mediated bridge and increased mobility of Wat795 (Fig. S6F) that affects binding entropy (Table 1), close contacts to the binding pocket Gly798-Val799 backbone (C1/C2, GTX2/3, and dcGTX2/3) (Fig. S7J), and an apparent clash of R2 modified toxins with a key STX-binding residue, *Np*Sxph Glu541 (Fig. S8). By contrast, modifications that face the solvent-exposed mouth of binding pocket, such as carbamoylsulfation (GTX5) or linker addition[13], are well tolerated. Together, these data highlight the tradeoff between the rigidity of the high-affinity the STX site[13,14] in which water has a key role in mediating toxin binding and the ability of *Rc*Sxph and *Np*Sxph to accommodate STX congener modifications.

Sxphs and Na$_V$s have divergent rank orders of STX congener affinities that suggest binding site differences beyond the convergent framework used to recognize the STX bis-guanidinium core[13,14]. The GTX5 sulfocarbamate is well-tolerated by Sxphs but not Na$_V$s, whereas Na$_V$s are better at accommodating the C11 sulfation of GTX2/3 than Sxphs (Tables S1–S3 and S5). Comparison of the Sxph and Na$_V$ toxin binding sites indicates that these differences are rooted in the divergent orientation of the toxins in the two binding sites that is set by the location of the core STX recognition elements. In Na$_V$s, the toxin sits laterally across the ring of the channel pore such that Domains I and II elements engage the bis-guanidinium core while DIII and DIV frame the carbamate and C11 sites (Fig. 7A, B). By contrast, Sxphs present a vase-like binding cavity in which the STX core binding residues are at the bottom of a deep pocket from which the carbamate and C11 sites project to the solvent exposed opening (Fig. 7C, D). The resulting orientation differences lead to sulfocarbamate clashes with Na$_V$

Domain III selectivity filter (Fig. 7A, B) that align with evidence for carbamate-DIII energetic coupling[36] and strong loss of binding (Table S3). Such clashes with carbamate modifications are absent in Sxphs (Fig. 7C, D). Indeed, Sxphs allow large modifications at this site with no consequence to affinity[13]. Conversely, in Na$_V$s, C11 sulfation appears to be positioned in a solvent exposed part of the channel Domain IV selectivity filter free from steric clashes (Fig. 7A, B), consistent with its modest reduction in Na$_V$ block (~ twofold, e.g., STX vs. GTX2/3 Table S5)[37]. This situation differs from Sxph where C11 sulfation yields close contacts with a binding pocket wall (Fig. 7C, D) and affects the stability of the water molecule that bridges the Asp795 and the toxin (Fig. S6F). These comparisons underscore the importance of toxin orientation with respect to the surrounding features of its binding site and the crucial role of water in mediating receptor-toxin interactions. Both factors are likely to be important in designing Sxphs or other proteins capable of recognizing various STX congeners. Whether STX congeners bind to Na$_V$s in orientations that differ from STX and whether water-mediated interactions like those seen in Sxph:toxin complexes are critical for high affinity Na$_V$ binding remain key undressed questions.

Naturally occurring PST mixtures comprising STX and its congeners are potent Na$_V$ blockers[1,3–6], making such STX variants the concern of public health efforts to mitigate the effects of PSP outbreaks. Deciphering how STX and its congeners interact with their molecular targets is important for developing ways to detect and neutralize PSTs, particularly as much seafood monitoring largely relies on *a* ~ 90 year old mouse lethality assay[39,40]. A major challenge is developing reagents that can capture and distinguish different STX congeners. In this regard, it is notable that Sxphs are very stable having Tms ->50 °C[13], produce toxin binding curves of comparable quality to rat brain homogenate in the RBA[17,23] (Fig. S2 and Table S3), and show a broader dynamic range in this assay. Hence, our results demonstrate that Sxphs can serve as an attractive, thermostable alternative for PST detection that merits further study.

Sxphs are thought to act as 'toxin sponges' that protect some frogs from STX poisoning by competing with Na$_V$s for toxin binding[15,18]. Our studies show that *Rc*Sxph and *Np*Sxph rescue both amphibian and human Na$_V$1.4 from STX congener block (Figs. 5 and 6). Similar to STX[13], the ability of these proteins to sequester toxin congeners is linked to Sxph:toxin affinity, as weaker Sxph binders such as GTX2/3 require higher Sxph:toxin ratios to affect rescue (Fig. 6D, E). Further, *Rc*Sxph and *Np*Sxph show distinct capacities to discriminate STX from dcSTX (Table S2) and different kinetic behaviors in GTX2/3 rescue assays (e.g., Fig. S7G–I). These two Sxphs vary at a key position that influences toxin binding affinity (Tyr558 and Ile559, respectively)[13], providing support for the idea that natural Sxph STX binding pocket variations shape STX congener selectivity[13]. NeoSTX series members comprise some of the most potent STX congeners[1,2,5,16] but do not have appreciable affinity for *Rc*Sxph or *Np*Sxph. Given Sxph STX binding pocket sequence diversity[13] and evidence for neoSTX binding activity in cane toad (*Rhinella marina*) plasma[21], an organism that has a Sxph[13], profiling the STX congener binding of diverse Sxphs could provide a path to identify Sxphs capable of neutralizing neoSTX series toxins. Together, our results establish the necessary foundation for understanding how Sxphs sequester diverse neurotoxins and highlight the key role of water in toxin binding. This framework is important for understanding toxin sponge resistance mechanisms[15,18] and for the development of means to detect and neutralize diverse types of PSTs.

## Methods

### Ethical statement
Oocytes were harvested from female *Xenopus laevis* frogs (National Xenopus Resource, Marine Biological Lab) and housed in the UCSF Laboratory Animal Resource Center (LARC) facilities. The use of these *Xenopus* oocytes was approved by IACUC (protocol approval # AN193390 - 01B) and experiments were performed in accordance with University of California guidelines and regulations.

### Sxph expression and purification
*Rana catesbeiana* saxiphilin (*Rc*Sxph) (GenBank: U05246.1) and *Nanorana parkeri* saxiphilin (*Np*Sxph) (GenBank: XM_018555331.1) were produced by expression in insect cells using baculoviruses and were purified as previously described[13,14,22]. In brief, Sxphs carrying a C-terminal 3 C protease cleavage site, green fluorescent protein (GFP), and a His$_{10}$ tag in series were expressed in *Spodoptera frugiperda* (*Sf*9) cells using a baculovirus expression system[22]. After addition of P2/P3 baculovirus to *Sf*9 cells at the dilution ratio of 1:50 (v/v), cells were incubated in a non-humidified New Brunswick Innova 44 incubator (Eppendorf, cat. no. M1282-0010) at 27 °C, shaking at 130 rpm for 72 h. Expressed Sxphs were secreted into the growth media. Cells were harvested by centrifugation. The supernatant was adjusted to pH 8.0 with a final concentration of 50 mM Tris-HCl and treated with 1 mM NiCl$_2$ and 5 mM CaCl$_2$ to precipitate contaminants. Precipitants were removed by centrifugation, and the clarified supernatant was incubated with antiGFP nanobody-conjugated Sepharose resin for 5 hours at room temperature (23 ± 2 °C). The resin was washed with 20 column volumes of a wash buffer containing 300 mM NaCl and 30 mM Tris-HCl (pH 7.4). After purification with antiGFP nanobody resin, protein samples were treated with 3 C protease (0.2 mg mL$^{-1}$ in the wash buffer) overnight at 4 °C to remove the GFP-His tag from Sxphs. The cleaved eluates were collected and purified by size exclusion chromatography (SEC) using a Superdex 200 10/300 GL column (Cytiva). For the TF, FP, RBA, ITC assays, and electrophysiology experiments, Sxphs were purified using a final SEC step in 150 mM NaCl,10 mM HEPES (pH 7.4). Structure determination of *Np*Sxph complexes with STX congeners, was done following the same methods, except for the final SEC buffer containing 30 mM NaCl, 10 mM HEPES (pH 7.4). Protein concentrations were determined by measuring UV absorbance at 280 nm using the following extinction coefficients calculated using the ExPASY server (https://web.expasy.org/protparam/): *Rc*Sxph and *Rc*Sxph E540A mutant 96,365 M$^{-1}$ cm$^{-1}$; *Np*Sxph 108,980 M$^{-1}$ cm$^{-1}$[22].

### Toxin preparation
Saxitoxin (STX) and fluorescein-labeled saxitoxin (F-STX) were synthesized, purified, and validated as outlined in refs. 13,22,41,42. STX and F-STX powders were directly dissolved with MilliQ water to make 1 mM and 1 μM stocks, respectively. Decarbamoylsaxitoxin (dcSTX, cat. no. dcSTX-c), gonyautoxin 2/3 (GTX2/3, cat. no. GTX2/3-d), decarbamoylgonyautoxin-2/3 (dcGTX2/3, cat. no. dcGTX2/3-d), gonyautoxin-5 (GTX5, cat. no. GTX5-d), N-sulfocarbamoylgonyautoxin-2/3 (C1/C2, cat. no. C1/C2-c), decarbamoylneosaxitoxin (dcneoSTX, cat. no. dcneoSTX-d), gonyautoxin-6 (GTX6, cat. no. GTX6-b), and gonyautoxin-1/4 (GTX1/4, cat. no GTX1/4-e) were purchased from the National Research Council Canada (NRC). Neosaxitoxin (neoSTX, cat. no. 41619) was purchased from Sigma-Aldrich. All the purchased toxins were lyophilized prior to making 1–5 mM stocks using MilliQ water.

### Thermofluor (TF) assay
Thermofluor assays[43] for the STX congeners (dcSTX, GTX2/3, dcGTX2/3, GTX5, C1/C2, neoSTX, dcneoSTX, GTX6 and GTX1/4) were performed as previously described for STX[13,22]. Briefly, twofold serial dilutions of each toxin were prepared in 150 mM NaCl, 10 mM HEPES, pH 7.4. 20 μL samples containing 1.1 μM *Rc*Sxph or *Np*Sxph, 5× SYPRO Orange dye (Sigma-Aldrich, cat. no. S5692, stock concentration 5000×), 0–20 μM toxin, 150 mM NaCl, 10 mM HEPES, pH 7.4 were set up in 96-well PCR plates (Bio-Rad, cat. no. MLL9601), sealed with a microseal B adhesive sealing film (Bio-Rad, cat. no. MSB1001) and centrifuged (1 min, 230 × *g*) prior to thermal denaturation using a CFX

Connect Thermal Cycler (Bio-Rad, cat. no. 1855201). Fluorescence was measured using the HEX channel (excitation $\lambda = 515-535$ nm, emission $\lambda = 560-580$ nm). Samples were incubated at 25 °C for 2 min followed by a temperature gradient from 25 °C to 95 °C at 0.2 °C min$^{-1}$, and final incubation at 95 °C for 1 min.

For the toxin dose-response curves, Sxph melting temperatures (Tms) in the presence of varied toxin concentrations were calculated by fitting the denaturation curves using a Boltzmann function in GraphPad Prism (GraphPad Software) using the equation $F = F_{min} + (F_{max}-F_{min})/(1+\exp((Tm-T)/C))$, where F is the fluorescence intensity at temperature T, $F_{min}$, and $F_{max}$ are the fluorescence intensities before and after the denaturation transition, respectively, Tm is the midpoint temperature of the transition, and C is the slope at Tm. $\Delta$Tms for Sxph in the absence ($Tm_{Sxph}$) and presence ($Tm_{Sxph+toxin}$) of different toxin concentrations were calculated using the following equation: $\Delta Tm = Tm_{Sxph+toxin}-Tm_{Sxph}$.

### Fluorescence polarization competition (FPc) assay

Fluorescence polarization competition (FPc) assays[28] were performed using 100 μL total reaction volume per well and final concentrations of the fluorescent ligand and Sxph of: for *Rc*Sxph, 1 nM F-STX and 12 nM *Rc*Sxph; for *Np*Sxph, 0.5 nM F-STX and 1.6 nM *Np*Sxph. For the displacement experiments, twofold serial dilutions of unlabeled toxins (STX, dcSTX, GTX2/3, dcGTX2/3, GTX5, C1/C2, neoSTX, dcneoSTX, GTX6, and GTX1/4) were prepared using in a buffer of 150 mM NaCl, 10 mM HEPES, pH 7.4 at the following concentration ranges: STX and GTX5, 0–875 nM; dcSTX and GTX2/3, dcGTX2/3, C1/C2, 0–7 μM; GTX6, 0–14 μM; and neoSTX, dcneoSTX, and GTX1/4, 0–28 μM. Samples containing 2 nM F-STX, 24 nM *Rc*Sxph, and 150 mM NaCl, 10 mM HEPES, pH 7.4 or 1 nM F-STX, 3.2 nM *Np*Sxph, and 150 mM NaCl, 10 mM HEPES, pH 7.4 were incubated for 1 h at room temperature (23 ± 2 °C) protected from light. Subsequently, 50 μL the equilibrated Sxph:F-STX solution was mixed with 50 μL of each toxin dilution series in 96-well black flat-bottomed polystyrene microplates (Greiner Bio-One, cat. no. 655900). The plate was sealed with an aluminum foil sealing film AlumaSeal II (Excel Scientific, cat. no. AF-100) and incubated at room temperature (23 ± 2 °C) for 2 h to attain equilibrium. Measurements were performed at 25 °C on a Synergy H1 microplate reader (BioTek) using the polarization filter setting (excitation $\lambda = 485$ nm, emission $\lambda = 528$ nm). Data were normalized using the following equation[28]: $P = (P_{Sxph:toxin}-P_{Toxin})/(P_{Ctrl}-P_{Toxin})$, where P is the polarization measured at a given toxin concentration, $P_{Sxph:toxin}$ is the polarization of Sxph:toxin mixture, $P_{Toxin}$ is the polarization of toxin in the absence of Sxph, and $P_{Ctrl}$ is the maximum polarization of Sxph in the absence of unlabeled toxin. The concentrations of the competing toxin causing displacement of 50% of bound F-STX ($IC_{50}$) were calculated by fitting normalized FP as a function of toxin concentration using the nonlinear regression analysis in GraphPad Prism (GraphPad Software).

### Radioligand receptor binding assay (RBA)

Radioligand RBA analysis for PSTs followed established protocols[17,23] with modifications. Tritiated saxitoxin ([3H]STX) was provided by American Radiolabeled Chemicals (St. Louis, MO; cat. no. ARK0101). Saxitoxin dihydrochloride (STX-diHCl, cat. no. NIST-8642a) was purchased from the National Institute of Standards and Technology. All other unlabeled toxins were Certified Reference Material purchased from the NRC Canada (cat. no. dcSTX-c, GTX2/3-d, dcGTX2/3-c, GTX5-d, C1/C2-c, dcneoSTX-d, GTX6-b, GTX1/4-d, neoSTX-d).

Assays were performed in a Multiscreen 96-well GF/B microtiter filtration plate (Millipore, cat. no. MSFBN6B) with 210 μL total per well of the following: 35 μL assay buffer (20 mM HEPES, 100 mM NaCl, 1 mM EDTA, pH 7.4), 35 μL unlabeled toxin, 35 μL [3H]STX, and 105 μL receptor solution (rat brain homogenate, *Np*Sxph, *Rc*Sxph, or *Rc*Sxph E540A). Serial dilutions of unlabeled toxins were prepared in either 0.003 N hydrochloric acid (Fisher Chemical) or 20 μM acetic acid (J.T.

Baker) based on the stock storage solution with the following in-well concentration ranges: STXdiHCl, dcSTX, GTX2/3, GTX5, GTX6, 0–1 μM/well; dcneoSTX, 0–5 μM; and GTX1/4, neoSTX, dcGTX2/3, C1/C2, 0–10 μM. The [3H]STX solution was prepared in cold assay buffer at a concentration of ~1 nM. Rat brain (Hilltop Lab Animals, Scottdale, PA) homogenate was prepared according to previously published protocols[17,23], then was diluted in cold assay buffer to a concentration of ~30 μg/well in assay. Saxiphilin proteins (*Np*Sxph, *Rc*Sxph, and *Rc*Sxph E540A) were prepared in cold assay buffer at a concentration of 0.14 μg/well (~$7.2 \times 10^{-3}$ μM/well). Following a 1 h incubation at 4 °C, wells were vacuum filtered and washed twice with 100 μL of cold assay buffer. Scintillation cocktail (Sigma Ultima Gold cat no. L8286; 50 μL/well) was then added and incubated at room temperature for 30 min. Radioactivity was measured (1 min/well) using a Perkin Elmer 2450 Microbeta microplate scintillation counter. Data were normalized using B/$B_{max}$, where B represents the bound [3H]STX in the sample and $B_{max}$ represents the maximum binding of [3H]STX in the absence of competing unlabeled STX. The half maximal effective concentrations ($EC_{50}$) were calculated by fitting normalized counts per minute as a function of toxin concentration using the nonlinear regression sigmoidal dose-response (variable slope) curve fit with weight by 1/$Y^2$ and bottoms constrained to a shared value for all data sets in GraphPad Prism.

### Isothermal titration calorimetry (ITC)

ITC measurements were performed at 25 °C using a MicroCal PEAQ-ITC calorimeter (Malvern Panalytical) as described previously[13,22]. *Np*Sxph was purified using a final SEC step in 150 mM NaCl, 10 mM HEPES, pH 7.4. 1 mM or 2 mM toxin stock in MilliQ water was diluted with the SEC buffer to prepare 100 μM or 200 μM toxin solutions having a final buffer composition of 135 mM NaCl, 9 mM HEPES, pH 7.4. To match buffers between the Sxph and toxin solutions, the purified protein samples were diluted to 10 μM or 20 μM with MilliQ water to reach a buffer concentration of 135 mM NaCl, 9 mM HEPES, pH 7.4. Protein samples were filtered through a 0.22 μm spin filter (Millipore, cat. no. UFC30GV00) before loading into the sample cell and titrated with toxin (100 μM for GTX5, dcSTX, GTX2/3 and C1/C2, and 200 μM for dcGTX2/3) using a schedule of 0.4 μL titrant injection followed by 35 injections of 1 μL for GTX5 and dcSTX, 18 injections of 2 μL for dcGTX2/3, and 24 injections of 1.5 μL for GTX2/3. The calorimetric experiment settings were: reference power, 5 μcal/s; spacing between injections, 150 s; stir speed 750 rpm; and feedback mode, high. Data analysis was performed using MicroCal PEAQ-ITC Analysis Software (Malvern Panalytical) using a fitted offset for correcting for the heat of dilution and the single binding site model.

### Crystallization and structure determination

*Np*Sxph was purified using a final SEC step in 30 mM NaCl, 10 mM HEPES, pH 7.4 as previously described[13], and concentrated to 30–40 mg mL$^{-1}$ using a 50-kDa cutoff Amicon Ultra centrifugal filter unit (Millipore, cat. no. UFC505096). Toxin stocks of dcSTX, GTX2/3, dcGTX2/3, GTX5, and C1/C2 for co-crystallization experiments were prepared at 5 mM. For co-crystallization of *Np*Sxph with STX congeners, *Np*Sxph was mixed with each toxin in a molar ratio of 1:1.2 *Np*Sxph:toxin. Samples were incubated on ice for 1 h before setting up crystallization trays. *Np*Sxph:dcSTX, *Np*Sxph:GTX2/3, *Np*Sxph:dcGTX2/3, *Np*Sxph:GTX5, *Np*Sxph:C1/C2 crystals were obtained by hanging drop vapor diffusion at 4 °C from 400 nl drops (1:1 (v/v) ratio of protein and precipitant) set with Mosquito crystal (SPT Labtech) using 20–25% (v/v) PEG 400, 4–5% (w/v) PGA-LM, 100–200 mM sodium acetate, pH 5.0. All *Np*Sxph:toxin crystals were harvested and flash-frozen in liquid nitrogen without additional cryoprotectant.

X-ray datasets for all *Np*Sxph:STX congener complexes were collected at 100 K at the Advanced Light Source 8.3.1 beamline (Berkeley,

CA), processed with XDS[44] and scaled and merged with Aimless[45]. The *Np*Sxph:toxin structures were solved by molecular replacement using the apo-*Np*Sxph structure (PDB:8D6G) as a search model in Phaser from PHENIX[46]. The electron density map and the model were manually checked in COOT[47] and iterative refinement was performed using phenix.refine[46]. The quality of all models was assessed using MolProbity[48] and refinement statistics.

## Molecular dynamics simulations

Atomic coordinates of *Np*Sxph structures with STX (PDB:8D6M)[13], dcSTX (PDB:8V68), GTX2 (PDB:8V69), dcGTX2 (PDB:8V65), GTX5 (PDB:8V66), and C1 (PDB:8V67), were used alongside those of crystallographic waters to prepare the corresponding topology and coordinate files using CHARMM-GUI[49]. Missing residues in the protein structures (Lys174, Arg175, Lys645, Ala646) were modeled using Prime (Schrödinger Suite 2024.1)[50]. Residues from the C-terminal 3 C protease cleavage sites (Ser827, Asn828, Ser829) were removed, leaving the C-termini negatively charged, while the N-termini were capped with an acetyl group. All residues were maintained in their dominant protonation states at pH 7.0, except for His71, His125, Glu211, Glu214, Asp267, His337, Asp392, and His425, which were protonated. Disulfide bonds were introduced between the following cysteine pairs: Cys10-Cys45, Cys20-Cys36, Cys27-Cys418, Cys91-Cys113, Cys124-Cys131, Cys133-Cys155, Cys163-Cys185, Cys196-Cys203, Cys205-Cys227, Cys235-Cys826, Cys259-Cys342, Cys304-Cys317, Cys314-Cys325, Cys370-Cys384, Cys477-Cys509, Cys487-Cys500, Cys534-Cys821, Cys552-Cys781, Cys589-Cys667, Cys623-Cys637, Cys634-Cys650, and Cys707-Cys721.

A water box, extending 10 Å from the protein surface, was built around the protein. Random water molecules were replaced with sodium and chlorine ions to neutralize the systems and achieve a physiological concentration of 0.121 M NaCl. The AMBER force field ff19SB[51] was employed for protein residue parameterization, and the OPC water mode l[52] was applied. Ligands were parameterized based on crystal structure coordinates using Sage (OpenFF 2.1.0)[53]. The topology and coordinate files for the system and ligands were manually combined, and overlapping water molecules were removed to generate the final systems. The final systems had dimensions of $122 \times 122 \times 122$ Å$^3$ and contained approximately 223,600 atoms, including 52,600 water molecules, 119 sodium ions, and 125 chloride ions.

Simulations were carried out using GROMACS 2024.2[54]. Each simulation system underwent energy minimization, followed by equilibration in the constant temperature, constant volume (NVT) ensemble for 1 ns, with harmonic restraints of 1.0 kcal·mol$^{-1}$·Å$^2$ on the ligand atoms and protein heavy atoms, and 0.1 kcal·mol$^{-1}$·Å$^{-2}$ on protein hydrogen atoms. Five independent 1-μs production runs were conducted for each toxin-bound *Np*Sxph complex in the constant temperature and constant pressure (NPT) ensemble. Temperature control was implemented with a velocity-rescaling thermostat[55], using a 1.0 ps time constant and a reference temperature of 303.15 K, while pressure control used a stochastic cell rescaling isotropic barostat[56] with a 5.0 ps time constant, a reference pressure of 1 bar, and a compressibility of $4.5 \times 10^{-5}$ bar$^{-1}$. The LINCS algorithm[57] was used to constrain the bond lengths of hydrogen atoms. Periodic boundary conditions were applied, with a 10.0 Å cutoff for Lennard-Jones and short-range electrostatic interactions. Long-range electrostatics were computed using the particle mesh Ewald[58] method with a fourth-order interpolation scheme and fast Fourier transform grid spacing of 1.25 Å. A continuum model correction was applied to account for long-range van der Waals interactions. The equations of motion were integrated with a 2 fs time step.

The root-mean-square deviation (RMSD) of the binding pose for each toxin over time was calculated across the five replicas for each system using MDTraj 1.9.4[59]. The initial 10 ns of simulation time from each trajectory were excluded from the analysis. For this analysis, snapshots in the trajectories were aligned based on the Cα atoms of the protein that were within 15 Å of the ligand's center of mass, considering all trajectories. The stability of the ligand core, assessed by the RMSD of atoms shared by all ligands relative to their initial structures, was compared across the different ligands. The conformational plasticity of the α6C1-β6C1 loop (residues Glu784-Asp791) was assessed by calculating the root mean square fluctuation (RMSF) of Cα atoms with MDTraj 1.9.4[59], with error bars calculated as the standard deviation of the RMSF over the 5 trajectories for each toxin-bound *Np*Sxph complex.

Structural interaction fingerprints for each toxin-bound *Np*Sxph complex were calculated using PLIP v2.3.0[60] with default settings. The initial analysis focused on identifying salt bridges, hydrogen bonds, water-bridged hydrogen bonds, π-cation interactions, hydrophobic interactions, and π-stacking interactions. However, due to the low occurrence of π-cation, hydrophobic, and π-stacking interactions (present in less than 0.1% of all frames for any ligand-bound *Np*Sxph complex), these were excluded from further analysis. To simplify the representation of the remaining interactions, ligand moieties were categorized by their functional groups: carbamate (specific to GTX5, STX, GTX2/3, and C1/2), C12 hydrate (common to all toxins), C11 sulfate (specific to GTX2/3, C1/C2, and dcGTX2/3) *N*-sulfate (specific to GTX5 and C1/C2) and the STX core (encompassing the remaining atoms in all ligands). The protein residues involved in interactions were grouped, and their contributions to the fraction of the frames where interactions were present were summed.

The occupancy of Wat541, Wat795, and Wat786 near residues Glu541, Asp795, and Asp786 respectively, was determined by calculating the minimum distance between the water molecules and the Cδ or Cγ atoms of Glu541 or Asp795, respectively. Due to the flexibility of the α6C1-β6C1 loop, the occupancy of Wat786 near Asp786 was only detected when the toxin N7 was within ~4.2 Å of the Cγ of Asp786. Therefore, it was calculated as a function of the minimum distance between the toxin N7 and the Cγ of Asp786. Subsequently, the water occupancy probability for each trajectory was assessed as the fraction of frames where the calculated minimum distances were less than 4.5 Å. The results are presented as mean values along with the 25% and 75% quantiles.

## GTX2/3 synthesis

The preparation of gonyautoxin 2/3 (GTX 2/3) follows previously published work[33,61]. Synthesis of this toxin relies on a key oxidative dearomatization of a pyrrole ring to form the bis-guanidine tricyclic core that is common to all paralytic shellfish poisons. Subsequent functional group interconversion steps enable installation of the C11-alcohol and C12-ketone groups. A final step sequence involving reductive deprotection to liberate the two protected guanidine groups and C11-alcohol sulfation delivers the natural product as a single isomer, gonyautoxin 3. However, upon standing in buffered solution, epimerization of the C11-sulfate occurs to give a 3:1 equilibrium mixture of GTX 2 and 3, consistent with the original isolation report. The structure of the synthetic material was validated by $^1$H NMR and mass spectrometry, and its potency was confirmed through whole-cell electrophysiology recordings against Na$_V$1.4 expressed in CHO cells.

## Automated patch-clamp electrophysiology

Chinese hamster ovary (CHO) cells stably expressing the α-subunit of human Na$_V$1.4 sodium channel (*Hs*Na$_V$1.4) (B'SYS GmbH, cat. no. BSYS-NaV1.4-CHO-C) were cultured at 37 °C in a humidified incubator with 5% CO$_2$ in complete culture medium containing: Ham's F-12 medium with GlutaMAX (Gibco, cat. no. 31765035) supplemented with 9% (v/v) heat inactivated fetal bovine serum (HI FBS) (Gibco, cat. no. 16140071), and antibiotics (0.9% (v/v) penicillin/streptomycin solution (Gibco, cat. no. 15-140-122) and 100 μg/mL Hygromycin B (Sigma-Aldrich, cat. no. 10843555001)).

Whole-cell patch-clamp experiments were performed using a QPatch Compact (Sophion Bioscience). The extracellular solution (ECS, saline) contained 2 mM CaCl$_2$, 1 mM MgCl$_2$, 10 mM HEPES, 4 mM KCl, 145 mM NaCl, 10 mM glucose (pH 7.4 with NaOH), and osmolarity adjusted to 305 mOsm/L with sucrose. The intracellular solution contained 140 mM CsF, 1 mM/5 mM ethylene glycol-bis(β-aminoethyl ether)-N,N,N′,N′-tetraacetic acid (EGTA)/CsOH, 10 mM HEPES, 10 mM NaCl (pH 7.3 with 3 M CsOH), and osmolarity adjusted to 320 mOsm/L with sucrose. All solutions were kept at room temperature (23 ± 2 °C) before application to the CHO-*Hs*Na$_V$1.4 cells. Cells were dissociated using Detachin (AMSBIO, cat. no. T100100) and kept in serum-free medium (Sigma-Aldrich, cat. no. C5467) supplemented with 25 mM HEPES and 0.04 mg/mL soybean trypsin inhibitor until recording. Cells were washed and resuspended in the ECS to reach a cell density of 4–6 × 10$^6$ cells/mL shortly prior application to the QPatch Compact.

Sodium currents were elicited by a 60 ms depolarization step from −120 mV to 0 mV, with a holding potential of −120 mV and a sweep-to-sweep interval duration of 1.64 s. All experiments were conducted at room temperature (23 ± 2 °C) using single-hole QPlates (Sophion Bioscience, cat. no. SB0201).

To determine toxin dose-response curves, toxin solutions were prepared in a 3-fold serial dilution series in ECS. Cumulative concentration-response experiments for STX, dcSTX, GTX2/3, dcGTX2/3, GTX5, and C1/C2 were performed by applying increasing toxin concentrations to each CHO-*Hs*Na$_V$1.4 cell. Peak currents were sampled at 10 kHz and filtered at 333 Hz using an 8$^{th}$ order Bessel filter with leak subtraction applied. IC$_{50}$ values for each toxin were calculated by fitting the dose-response curves (normalized peak current ($I_x$/$I_0$) as a function of toxin concentration) using the following equation: $I_x$/$I_0$ = $(I_{max}-I_{min})$/$(1+x/IC_{50})$, where $I_x$ is the current amplitude at the toxin concentration $x$, $I_0$ is the current amplitude in the latest saline period before toxin application, and $I_{max}$ and $I_{min}$ are the maximum and minimum peak current amplitudes, respectively, and IC$_{50}$ is the half-maximal inhibitory concentration.

For the toxin neutralization experiment, *Np*Sxph protein stock solution a buffer of 150 mM NaCl and 10 mM HEPES, pH 7.4 was diluted in ECS to 25 μM for the experiments with C1/C2 and GTX5, and to 5 μM for experiments with STX, dcSTX, GTX2/3, and dcGTX2/3. *Np*Sxph was incubated with each toxin using 5:1 [*Np*Sxph]:[toxin] molar ratio for at least 30 min prior application to the CHO-*Hs*Na$_V$1.4 cells. Toxin solutions and *Np*Sxph:toxin mixture were prepared in ECS. Baseline sodium currents were first recorded in saline, and then using toxin concentrations sufficient to block ~90% peak current as established from the dose-response experiments: 50 nM for STX, 200 nM for dcSTX, 60 nM for GTX2/3, 300 nM for dcGTX2/3, 1 μM for C1/C2, and 2.7 μM for GTX5. Following toxin wash-out, *Np*Sxph:toxin mixture was applied to assess the effect of *Np*Sxph on the toxin response. Following the *Np*Sxph:toxin wash-out, toxin at the desired concentration (50 nM for STX, 200 nM for dcSTX, 900 nM for GTX2/3, 300 nM for dcGTX2/3, 1 μM for C1/C2, and 2.7 μM for GTX5) was applied to assess channel block. Peak currents were sampled at 25 kHz and filtered at 8333 Hz using an 8th order Bessel filter with leak subtraction applied. Normalized current was determined by using the following equation: I = $(I_{Sxph:toxin}-I_{Toxin})$/$(I_{ctrl}-I_{Toxin})$, where $I_{Sxph:toxin}$ is the current after application of *Np*Sxph:toxin mixture, $I_{Toxin}$ is the current after toxin application, and $I_{ctrl}$ is the basal current recorded in saline.

All data analyses were performed using the Sophion Analyzer Software (Sophion Bioscience) and GraphPad Prism (GraphPad Software).

## Two-electrode voltage clamp
TEVC recordings were performed as described previously[18]. In brief, a pcDNA3.1+ vector containing *Phyllobates terribilis* SCN4A (*Pt*Na$_V$1.4) DNA (GenBank: MZ545381.1)[18] was linearized with *Xba*I, and cRNA subsequently synthesized using mMESSAGE mMACHINE T7

Transcription Kit (Invitrogen, CA, USA). *Xenopus laevis* oocytes (harvested under UCSF IACUC protocol AN193390-01I) were injected with 4–8 ng *Pt*Na$_V$1.4 and recordings were performed 1–2 days post injection at room temperature (23 ± 2 °C). Oocytes were impaled with borosilicate recording microelectrodes (0.3–2.0 MΩ resistance), backfilled with 3 M KCl. Sodium currents were recorded using ND96 solution containing the following: 96 mM NaCl; 1 mM CaCl$_2$; 1 mM MgCl$_2$; 2 mM KCl; 5 mM HEPES (pH 7.5 with NaOH). Data was acquired using an Axoclamp 900B amplifier (Molecular Devices, CA, USA) controlled by pClamp software (v10.9, Molecular Devices), digitized at 10 kHz using an Axoclamp 1550B digitizer (Molecular Devices).

The effects of toxins and Sxph-toxin combinations on *Pt*Na$_V$1.4 were assessed using a 60-ms depolarization step from −120 to 0 mV, with holding potential of −120 mV and sweep-to-sweep duration of 10 s. Leak currents were subtracted using a P/4 protocol during data acquisition. The half maximal inhibitory concentration (IC$_{50}$) values for STX congeners against *Pt*Na$_V$1.4 were determined by perfusing increasing concentrations of toxin in series and calculated using nonlinear regression analysis in Prism (v10.1, GraphPad, MA, USA). Toxin concentrations required to inhibit 90% of the current (IC$_{90}$) were calculated from the determined IC$_{50}$ and Hill slope ($H$) according to the following equation: $IC_x = \left(\frac{x}{100-x}\right)^{1/H} IC_{50}$.

Sxph rescue experiments were conducted by first recording baseline currents in ND96, then perfusing sufficient toxin to give ~90% block (100 nM STX, 200 nM GTX2/3, or 800 nM dcSTX). Sxph was then applied directly into the 1-mL recording chamber. For all saxiphilin:toxin ratios, the concentration of Sxph stock solution was adjusted such that the added Sxph solution was less than 1% of the total recording chamber volume. Rescue responses were normalized according to the following equation: $I_{norm} = \frac{(I_x - I_{min})}{(I_{max}-I_{min})}$, where $I_x$ is the current following addition of Sxph at ratio $x$, $I_{min}$ is the current following toxin block, and $I_{max}$ was the maximal current observed in recording solution alone. Toxin rescue kinetics were modelled using non-linear regression of normalized currents over time, and quality of model fit was assessed using Akaike's Information Criterion, corrected for small sample size (AICc). Data analysis was performed using Clampfit (v11.0, Molecular Devices) and Prism (v10.1, GraphPad).

Oocytes were harvested from female *Xenopus laevis* frogs (National Xenopus Resource, Marine Biological Lab) and housed in the UCSF LARC facilities. The use of these *Xenopus* oocytes was approved by IACUC (protocol approval # AN193390 - 01B) and experiments were performed in accordance with University of California guidelines and regulations.

### Reporting summary
Further information on research design is available in the Nature Portfolio Reporting Summary linked to this article.

## Data availability
The data that support this study are available from the corresponding authors upon request. Coordinates and structure factors have been deposited in the Protein Data Bank (PDB) and for 8V68 (*Np*Sxph:dcSTX), 8V69 (*Np*Sxph:GTX2), 8V65 (*Np*Sxph:dcGTX2), 8V66 (*Np*Sxph:GTX5), and 8V67 (*Np*Sxph:C1). The coordinates for the first and last frames from MD simulations have been deposited in Zenodo [https://doi.org/10.5281/zenodo.15160575]. The source data underlying Figs. 1B–E, 2A–I, 3F, 5A–G, 6A–E, and Supplementary Figs. S1A–F, S2A–K, S6A–C, E, F, and S7A–I are provided as a Source Data file. Source data for previously published PDB codes are: 8D6G, 8D6M, and 6J8G. Source data are provided with this paper.

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

## Acknowledgements

We thank M. Huynh and A. Bara for technical help and K. Brejc for comments on the manuscript. This work was supported by grants DoD HDTRA-1-19-1-0040, HDTRA-1-21-1-0011, and HDTRA-1-23-1-0026 to D.L.M., and NIH-NIGMS R01-GM117263-05 to J.D. E.N. was supported by a fellowship from the Studienstiftung des deutschen Volkes and acknowledges the use of computational resources provided in part by the Erlangen National High Performance Computing Center (NHR@FAU) through the b132dc grant. Additional computations utilized resources from Scientific Computing at the Icahn School of Medicine at Mount Sinai, supported by the Clinical and Translational Science Awards (CTSA) grant UL1TR004419 from the National Center for Advancing Translational Sciences, as well as the Office of Research Infrastructure at the National Institutes of Health under award numbers S10OD026880 and S10OD030463. This research was supported in part by an appointment to the NOAA Research Participation Program administered by the Oak Ridge Institute for Science and Education (ORISE) through an interagency agreement between the U.S. Department of Energy (DOE) and NOAA.

## Author contributions

S.Z., S.A.N., Z.C., J.D., and D.L.M. conceived the study and designed the experiments. S.Z. and Z.C. performed the TF assay and produced and purified Sxphs. S.Z. established the FPc assay, performed the ITC studies, and determined the crystal structures of *Np*Sxph complexes. K.M.A. and T.A.L. designed and conducted the RBA experiments. E.N. designed, ran, and analyzed the molecular dynamics simulations. K.K. and D.P. contributed additional analyses of the simulations. S.A.N. performed two-electrode voltage clamp electrophysiology experiments. S.Z. and S.A.N. performed whole-cell patch clamp toxin concentration-response experiments. S.Z. established and performed human Na$_V$ neutralization experiments. H.S.H., E.R.P., and J.V.M. synthesized and quantified samples of F-STX, STX, and GTX2/3. S.Z., S.A.N., Z.C., K.M.A., E.N., K.K., D.P. M.F., and D.L.M. analyzed data. T.A.L., M.F., J.D., and D.L.M. provided guidance and support. S.Z., S.A.N., J.D., and D.L.M. wrote the paper.

## Competing interests

J.D. is a cofounder and holds equity shares in SiteOne Therapeutics, Inc., a start-up company interested in developing subtype-selective modulators of sodium channels. The scientific results and conclusions, as well as any views or opinions expressed herein, are those of the authors and do not necessarily reflect the policies and views of the Department of Commerce (DOC), NOAA, DOE, or ORAU/ORISE. The other authors declare no competing interests.
