## [Transparent Peer Review file · Nature Communications]

Structural basis for saxitoxin congener binding and neutralization by anuran saxiphilins

Corresponding Author: Professor Daniel Minor

Version 0:

Reviewer comments:

Reviewer #1

(Remarks to the Author)

The manuscript presents a novel and significant contribution, tackling an interesting biological question with potentially broad implications. This work uses advanced methodologies, which collectively offer a deeper understanding of saxitoxin binding and neutralization by anuran saxiphilins. The manuscript situates itself well within the context of established literature, referencing foundational studies while identifying unanswered questions. The results are meticulously presented, with comprehensive statistical analyses that are appropriately applied and explained. The figures and tables complement the narrative, providing clear visual summaries of the findings. Furthermore, the discussion integrates the results into the context of existing literature, highlighting the novelty and significance of the work while acknowledging its limitations in a balanced manner. The presentation of figures and tables is generally strong but could be optimized for greater clarity and impact. For instance, multiple figures present many data, thereby compromising readability. Expanding on the broader implications of the findings, would also strengthen the manuscript's impact and contextualization.

Reviewer #2

(Remarks to the Author)

Zakrzewska et al. studied interaction of different saxitoxin (STX) congeners with saxiphilins (Sxphs) from American bullfrog (*Rana catesbeiana*) RcSxph and High Himalaya frog (*Nanorana parkeri*) NpSxph versus voltage-gated sodium channels PtNaV1.4 and HsNaV1.4. The authors used an impressive panel of methods, including thermofluor (TF) assay, fluorescence polarization competition (FPc) assay, radioligand receptor binding assay (RBA), isothermal titration calorimetry (ITC), X-ray crystallography, two-electrode voltage clamp (TEVC) and planar whole-cell patch-clamp recordings, and molecular dynamics (MD) simulations to decipher the molecular mechanisms of STX congener interaction with RcSxph and NpSxph versus NaV1.4. In particular, Zakrzewska et al. determined how the most prevalent STX congener alterations (decarbamylation and carbamate sulfation at R1, N1 hydroxylation at R2, and C11 sulfation at R3) tune their affinity towards saxiphilins in the broad range of ~3 orders of magnitude and differently affect their binding to sodium channels. The identified molecular elements that play the key roles in STX congener-Sxph interaction explain the toxin sponge resistance mechanisms and lay foundations for the development of new molecules to detect and neutralize diverse types of paralytic shellfish toxins (PSTs).

The manuscript is very well written, the experimental results appear to be of high quality, the interpretations are just and fair. I fully support the publication of this manuscript in its current form and only have a couple of minor suggestions:

1. Line 267. The mentioned panel H is missing in Fig. S6.
2. Chemical structures in Figs. 1B and C, 3A-E, 5A-B,D,F-G, 6A,C are too small. Please consider increasing their size.

Reviewer #3

(Remarks to the Author)

This manuscript investigates how saxiphilins (Sxphs) from amphibians interact with saxitoxin (STX) and its derivatives through their high-affinity binding pockets, revealing the molecular mechanisms by which Sxphs act as "toxin sponges" and neutralize various STX congeners. Using techniques such as thermofluor (TF), fluorescence polarization (FP), isothermal titration calorimetry (ITC), X-ray crystallography, and molecular dynamics simulations, the authors found that the binding pocket of Sxphs is pre-organized to recognize and bind multiple STX congeners via a "lock-and-key" mechanism, despite affinity differences caused by toxin modifications. Crucial water molecules play key roles in the binding process, and the

balance between the rigidity and flexibility of the binding pocket enables Sxphs to accommodate diverse toxin modifications. Electrophysiological experiments further showed that Sxphs can effectively reverse the blockage of voltage-gated sodium channels (Navs) by STX congeners, providing a foundation for the development of new tools for PSP detection and neutralization strategies.

Overall, this manuscript provides a clear and valuable investigation into the interactions between saxiphilins and saxitoxin congeners, supported by robust experimental data. With minor revisions, it would be well-suited for publication.

1. Although the article provides a detailed description of the binding mechanisms between Sxphs and STX derivatives, the discussion section could further explore how these mechanisms impact practical applications, such as their potential use in PSP detection and neutralization strategies.

2. I have noticed that the numbering of key water molecules seems to follow the residue numbering of the interacting residues. Are these water molecules fixed, or is there a dynamic exchange process of water molecules involved?

Version 1:

Reviewer comments:

Reviewer #1

(Remarks to the Author)

Great work. My comments were addressed. Publish without delay.

Reviewer #2

(Remarks to the Author)

The authors addressed all my original comments and I have no further comments.

Reviewer #3

(Remarks to the Author)

The authors have addressed all previously raised concerns adequately. I recommend its publication in the current form.

We thank the reviewers for their critical comments and positive assessment of our work. Our responses are highlighted in blue text.

Reviewer #1 (Remarks to the Author):

The manuscript presents a novel and significant contribution, tackling an interesting biological question with potentially broad implications. This work uses advanced methodologies, which collectively offer a deeper understanding of saxitoxin binding and neutralization by anuran saxiphilins. The manuscript situates itself well within the context of established literature, referencing foundational studies while identifying unanswered questions. The results are meticulously presented, with comprehensive statistical analyses that are appropriately applied and explained. The figures and tables complement the narrative, providing clear visual summaries of the findings. Furthermore, the discussion integrates the results into the context of existing literature, highlighting the novelty and significance of the work while acknowledging its limitations in a balanced manner. The presentation of figures and tables is generally strong but could be optimized for greater clarity and impact. For instance, multiple figures present many data, thereby compromising readability. Expanding on the broader implications of the findings, would also strengthen the manuscript's impact and contextualization.

We respectfully disagree with the reviewer regarding the amount of data presented in the figures. We think this is essential to show as much supporting evidence for our claims as possible. Hence, we have not altered the content of any figures. Following the suggestion of Reviewer #2, we increased the size of the chemical structures to improve readability.

The last sentence of the abstract and manuscript both address the general implications for toxin detection and antidote development. We also discuss potential paths for finding Sxphs that could bind other important STX congeners. We have not elaborated further as we think that our statements are sufficient to set a general direction given the current state of knowledge for this class of toxin sponge proteins.

Reviewer #2 (Remarks to the Author):

Zakrzewska et al. studied interaction of different saxitoxin (STX) congeners with saxiphilins (Sxphs) from American bullfrog (*Rana catesbeiana*) RcSxph and High Himalaya frog (*Nanorana parkeri*) NpSxph versus voltage-gated sodium channels PtNaV1.4 and HsNaV1.4. The authors used an impressive panel of methods, including thermofluor (TF) assay, fluorescence polarization competition (FPc) assay, radioligand receptor binding assay (RBA), isothermal titration calorimetry (ITC), X-ray crystallography, two-electrode voltage clamp (TEVC) and planar whole-cell patch-clamp recordings, and molecular dynamics (MD) simulations to decipher the molecular mechanisms of STX congener interaction with RcSxph and NpSxph versus NaV1.4. In particular, Zakrzewska et al. determined how the most prevalent STX congener alterations (decarbamylation and carbamate sulfation at R1, N1 hydroxylation at R2, and C11 sulfation at R3) tune their affinity towards saxiphilins in the broad range of ~3 orders of magnitude and differently affect their binding to sodium channels. The identified molecular elements that play the key roles in STX congener-Sxph interaction explain the toxin sponge resistance mechanisms and lay foundations for the development of new molecules to detect and neutralize diverse types of paralytic shellfish toxins (PSTs).

The manuscript is very well written, the experimental results appear to be of high quality, the interpretations are just and fair. I fully support the publication of this manuscript in its current form and only have a couple of minor suggestions:

1. Line 267. The mentioned panel H is missing in Fig. S6.

This should have been a reference to Fig. S6F. It has been corrected.

2. Chemical structures in Figs. 1B and C, 3A-E, 5A-B,D,F-G, 6A,C are too small. Please consider increasing their size

We have revised Figures 1B and C, 3A-E, 5 A-B, D, F-G, and 6A, C. Although not requested, we also revised Supplementary Figures S1A and B and S7 A-I as they had similar chemical structure icons. We have increased the size of the chemical structures in these figures as suggested.

Reviewer #3 (Remarks to the Author):

This manuscript investigates how saxiphilins (Sxphs) from amphibians interact with saxitoxin (STX) and its derivatives through their high-affinity binding pockets, revealing the molecular mechanisms by which Sxphs act as "toxin sponges" and neutralize various STX congeners. Using techniques such as thermofluor (TF), fluorescence polarization (FP), isothermal titration calorimetry (ITC), X-ray crystallography, and molecular dynamics simulations, the authors found that the binding pocket of Sxphs is pre-organized to recognize and bind multiple STX congeners via a "lock-and-key" mechanism, despite affinity differences caused by toxin modifications. Crucial water molecules play key roles in the binding process, and the balance between the rigidity and flexibility of the binding pocket enables Sxphs to accommodate diverse toxin modifications. Electrophysiological experiments further showed that Sxphs can effectively reverse the blockage of voltage-gated sodium channels (NaVs) by STX congeners, providing a foundation for the development of new tools for PSP detection and neutralization strategies.

Overall, this manuscript provides a clear and valuable investigation into the interactions between saxiphilins and saxitoxin congeners, supported by robust experimental data. With minor revisions, it would be well-suited for publication.

1. Although the article provides a detailed description of the binding mechanisms between Sxphs and STX derivatives, the discussion section could further explore how these mechanisms impact practical applications, such as their potential use in PSP detection and neutralization strategies.

Please note our response to Reviewer #1 who raised a similar point. As these issues are addressed in both the abstract and 'Discussion' section, we have not elaborated further.

2. I have noticed that the numbering of key water molecules seems to follow the residue numbering of the interacting residues. Are these water molecules fixed, or is there a dynamic exchange process of water molecules involved?

These residues are numbered based on the observation of their positions in our crystal structures and are named for the interacting residues. To clarify their naming further, we have added the following on p. 8:

'The high resolution structures also revealed a set of water-mediated networks that contribute to *NpSxph*:toxin interactions that we identify based on the residue with which they interact.'

The behavior of these waters is analyzed in our Molecular Dynamics analysis (p. 9). This work (Figs. S6E-F) shows that the key waters of interest (Wat541 and Wat795). By contrast, Wat786, which has lower occupancy in the crystal structures, is more mobile. The mobility, Wat795, is affected by C11 sulfation in STX congeners.

To clarify further we added the following on p. 9 (new text in blue):

'This Wat795 behavior agrees with the observation that C11 sulfated congeners show a more favorable binding entropy (Table 1) and supports the idea that toxin and binding pocket solvation changes contribute to toxin affinity. Hence, the mobility of this key water molecule is influenced by the STX congener identity.'

So that this point is not missed by the reader, we also added the following in the 'Discussion' (p. 14), new text in blue:

'These affinity differences arise from loss of carbamate contacts¹⁰ (dcSTX and dcGTX2/3), destabilization of the Asp795 water-mediated bridge and increased mobility of Wat795 (Fig.S6F) that affects binding entropy (Table 1), close contacts to the binding pocket Gly798-Val799 backbone (C1/C2, GTX2/3, and dcGTX2/3) (Fig. S7J), and an apparent clash of R2 modified toxins with a key STX-binding residue, *NpSxph* Glu541 (Fig. S8).'

Reviewer's comments are below. There were no remaining issues to address.

REVIEWERS' COMMENTS

Reviewer #1 (Remarks to the Author):

Great work. My comments were addressed. Publish without delay.

Reviewer #2 (Remarks to the Author):

The authors addressed all my original comments and I have no further comments.

Reviewer #3 (Remarks to the Author):

The authors have addressed all previously raised concerns adequately. I recommend its publication in the current form.